

# A regionally resolved inventory of High Mountain Asia surge-type glaciers, derived from a multi-factor remote sensing approach

**Gregoire Guillet**[1], **Owen King**[1], **Mingyang Lv**[3], **Sajid Ghuffar**[1,2], **Douglas Benn**[1], **Duncan Quincey**[4], **and Tobias Bolch**[1]

[1]School of Geography and Sustainable Development, University of St Andrews, United Kingdom
[2]Department of Space Science, Institute of Space Technology, Islamabad, Pakistan
[3]Aerospace Information Research Institute, Chinese Academy of Sciences, Beijing, 100094, China
[4]School of Geography, University of Leeds, United Kingdom

**Correspondence:** Gregoire Guillet (gg70@st-andrews.ac.uk)

**Abstract.** Knowledge about the occurrence and characteristics of surge-type glaciers is crucial due to the impact of surging on glacier melt and glacier-related hazards. One of the super-clusters of surge-type glaciers is High Mountain Asia (HMA). However, no consistent region-wide inventory of surge-type glaciers in HMA exists. We present a regionally resolved inventory of surge-type glaciers based on their behaviour across High Mountain Asia between 2000 and 2018. We identify surge-type behaviour from surface velocity, elevation and feature change patterns using a multi-factor remote sensing approach that combines yearly ITS_LIVE velocity data, DEM differences and very-high-resolution imagery (Bing Maps, Google Earth). Out of the ≈ 95 000 glaciers in HMA, we identified 666 that show diagnostic surge-type glacier behaviour between 2000 and 2018 is mainly found in the Karakoram (223) and the Pamir regions (223) CEI. The total area covered by the 666 surge-type glaciers represents 19.5 % of the glacierized area in Randolph Glacier Inventory (RGI) V6.0 polygons in HMA. A total of 68 glaciers were already identified as "surge type" in the RGI V6.0. We further validate 107 glaciers previously labelled as "probably surge type" and newly identify 491 glaciers, not previously reported in other inventories covering HMA. We finally discuss the possibility of self-organized criticality in glacier surges. Across all regions of HMA, the surge-affected area within glacier complexes displays a significant power law dependency with glacier length.

# 1 Introduction

Glacier surges are internally triggered, quasi-periodic oscillations of a glacier's dynamical behaviour, alternating between slow and fast flow (Meier and Post, 1969; Raymond, 1987; Sharp, 1988; Truffer et al., 2021). While periods of slow flow (or quiescence) usually last for tens to hundreds of years, periods of fast flow (surge) are shorter (months to 15 years), with flow velocity reaching up to 10–1000 times its standard order of magnitude (Meier and Post, 1969; Murray et al., 2003; Pritchard et al., 2005; Mansell et al., 2012). During surges, a substantial volume of ice is transferred from a reservoir zone down-glacier to a receiving zone. This mass transfer typically leads to drastic thinning in the upper reaches of the glacier and thickening at lower elevations, often causing a substantial advance of the glacier terminus (see Sund et al., 2009, for example).

Glacier surges can occur on both polythermal and temperate glaciers. In the former case, the bed can oscillate between cold and warm states over the course of a surge cycle, whereas in the latter the bed remains warm throughout. These two cases have been used as the basis for a two-fold classification of surge-type glaciers (Svalbard type vs. Alaskan type) with distinct instability mechanisms (thermal switch and hydrological switch, respectively) (Murray et al., 2003; Jiskoot et al., 2001). This binary view, however, reflects neither the wide diversity of surging behaviour (see Quincey et al., 2015, for example) nor the consistent associations between surging and climate, glacier geometry and substrate characteristics. Benn et al. (2019) have argued that the full

spectrum of glacier dynamic behaviour can be understood in terms of basal enthalpy balance. That is, stable steady states (non-surging behaviour) are only possible if enthalpy (thermal energy and water) generated at glacier beds from geothermal heat, friction and other sources can be evacuated from the system at the same rate. Imbalances between enthalpy inputs and outputs result in dynamic instabilities, including surges. Some combinations of regional (climatic) and local (topographic, geologic) factors encourage instability, consistent with observations (Sevestre and Benn, 2015). Many aspects of enthalpy balance theory need to be worked out in detail (e.g. interactions between ice motion, friction and basal drainage), and predictions of the theory need to be tested against local and regional datasets.

Surge-type behaviour has been documented for around 1 % of glaciers worldwide (Jiskoot et al., 1998; Sevestre and Benn, 2015), with two main super-clusters: High Mountain Asia (HMA) and the so-called "Arctic ring" (Alaska, Arctic Canada, Svalbard, and Russian high Arctic) (Sevestre and Benn, 2015). Surge-type glaciers throughout HMA have received significant attention in the past decades, with many regional inventories being generated, especially in the Karakoram (Hewitt, 1969; Kotlyakov et al., 1997; Barrand and Murray, 2006; Quincey et al., 2011; Copland et al., 2011; Bolch et al., 2017; Bhambri et al., 2017), Pamirs (Osipova et al., 1998; Kotlyakov et al., 2008; Shangguan et al., 2016; Lv et al., 2019; Goerlich et al., 2020) and Tien Shan (Osmonov et al., 2013; Mukherjee et al., 2017; Zhou et al., 2021). More recently, glacier surges have been documented in understudied regions of HMA such as the Tibetan Plateau (King et al., 2021; Xu et al., 2021; Zhu et al., 2021) and Western Kunlun Shan (Yasuda and Furuya, 2013, 2015; Chudley and Willis, 2019; Muhammad and Tian, 2020). Many of these studies have focused on the modern satellite era and highlighted the non-uniform distribution of surge-type glaciers in HMA, giving a more accurate representation of the prevalence of surge-type glaciers compared to ground-based efforts (Dolgoushin and Osipova, 1975; Kotlyakov et al., 2008; Imran and Ahmad, 2021). The only existing inventory of surge-type glaciers covering HMA in its entirety originates from the study of Sevestre and Benn (2015). Their inventory however relies on a compilation of scientific literature, albeit considering inconsistent diagnostic criteria, identification methods and study periods (1861–2013).

The cyclical advance of surge-type glaciers can result in repeated and widespread glacier hazard (i.e. outburst of glacier-dammed lakes, gravitational instabilities, etc.) formation (Gardner and Hewitt, 1990; Ding et al., 2018; Bhambri et al., 2019; Truffer et al., 2021). Hazards associated with surge-type glaciers impact both the local environment in the path of the advancing glacier terminus (Shangguan et al., 2016) and communities further downstream, in the path of meltwater produced by glaciers at the head of mountain catchments. The source of many large glacial lake outburst floods (GLOFs) in the Karakoram can be traced back to lakes which have formed behind the advancing terminus of a surge-type glacier (Hewitt and Liu, 2010; Bhambri et al., 2019). The timing and magnitude of the drainage of such lakes is difficult to predict because of the active nature of the lake's ice dam, but several GLOFs may occur from the same lake during the active phase of the damming glacier (Round et al., 2017; Gao et al., 2021; Bazai et al., 2021). The recent surge and associated glacial lake formation alongside Shishper Glacier (Bhambri et al., 2020; Muhammad et al., 2021) is just one example of the threat posed by surge-related hazards to high-mountain infrastructure such as the international Karakoram Highway – a route which provides a vital link to mountain communities in Tibet/southwest China and Pakistan (Ding et al., 2018).

The impact of ice redistribution through a glacier surge's cycle on its overall rate of mass loss or gain (mass balance) over multi-decadal timescales is still disputed. The flux of ice from a surge-type glacier's reservoir zone to its receiving zone results in the redistribution of a large ice volume to lower altitude, where it is more prone to ablation (Sund et al., 2009; King et al., 2021). Studies have documented increased melt rate in the expanded ablation zone, outpacing the rate of quiescent phase ice build-up in the years following surge cessation and leading to markedly more negative mass balance (Aðalgeirsdóttir et al., 2005; Kochtitzky et al., 2019; King et al., 2021). Similarly, synchronous surging by a number of glaciers in the Ak-Shirak region of the central Tien Shan resulted in enhanced ice loss in the 1980s and 1990s when no significant change in temperature or precipitation otherwise occurred to drive enhanced ablation (Bhattacharya et al., 2021). The surge-related fluctuation of individual glacier mass balance does not appear pronounced enough to influence regional ice loss rates over shorter timescales amongst the larger surge-type glacier clusters, such as the Karakoram (Bolch et al., 2017; Gardelle et al., 2013; Berthier and Brun, 2019). A detailed inventory of surge-type glaciers is clearly important for a region such as HMA, where glacier hazard mitigation and glacier meltwater yield will be a high priority as the temperatures of the region are projected to further rise in coming decades (Kraaijenbrink et al., 2017; Bolch et al., 2019; Lalande et al., 2021).

The objectives of this paper are twofold. First, we aim to assess the occurrence of surge-type glaciers across HMA using remotely sensed data spanning the period 2000–2018. Then, we aim to clarify what are the main geometric characteristics and controls for surging in HMA, as well as providing a clearer image of the spectrum of possible surge behaviour, in light of the enthalpy balance theory (Benn et al., 2019).

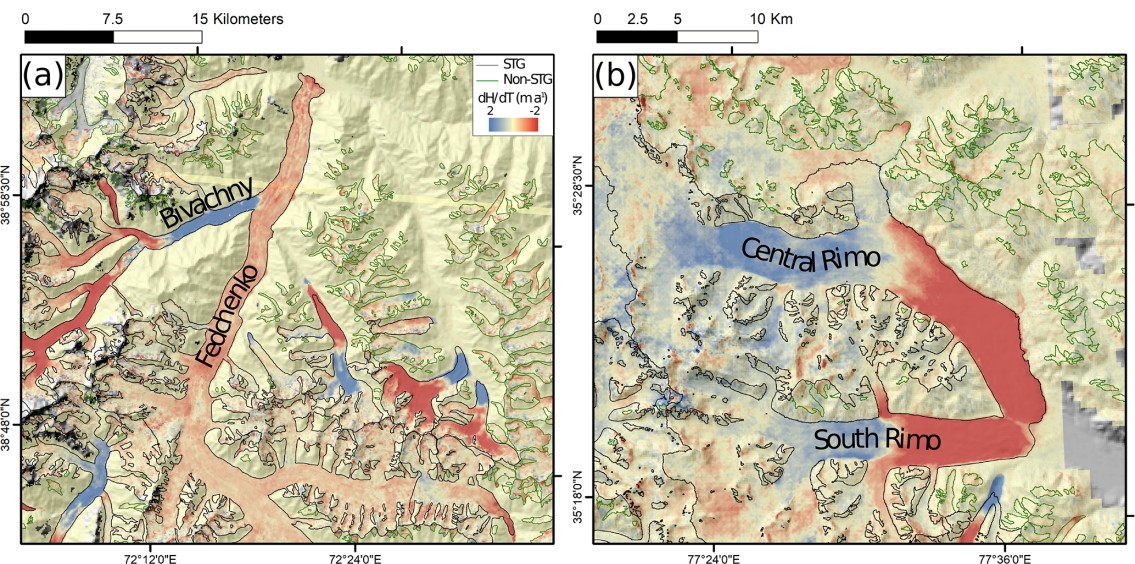

**Figure 1.** Examples of surface elevation change data used to identify surge-type glaciers. **(a)** Ice mass transfer associated with the surges of Bivachny Glacier, and other local glaciers, over the period 2010–2019 (data from Hugonnet et al., 2021). **(b)** Ice mass build-up in the reservoir zones of Central and South Rimo glaciers over the period 2000–2018 (data from Shean et al., 2020). Glacier outlines are from the Randolph Glacier Inventory V6.0 (RGI V6.0) (RGI Consortium, 2017).

## 2 Data and methods for surge-type glacier identification

We propose to identify surge-type glaciers from distinct widely used criteria (Dowdeswell and Williams, 1997; Copland et al., 2003; Grant et al., 2009), directly contrasting with regional trends of glacier mass loss (Brun et al., 2017; Shean et al., 2020; Bhattacharya et al., 2021), slowdown (Dehecq et al., 2019) and retreat (Li et al., 2019). First, substantial and spatially concentrated surface elevation changes (over 1–10 years), either in the reservoir zone or at lower elevations (near the glacier terminus), are assumed to be typical of surge-type glacier ice mass redistribution. Then, substantial variations in a glacier's velocity field over a similar time period are taken as indicative of surging. Finally, we consider chaotic glacier-wide crevasse patterns mixing longitudinal and transverse crevasses at low altitudes (typically close to the front) as diagnostic of active glacier surges. While terminus advance is often a consequence of glacier surges, not all surge-type glaciers display terminal advance during the active phase, and it is therefore not considered as a discriminating criterion for surge identification (Mukherjee et al., 2017; Paul et al., 2017; Steiner et al., 2018).

### 2.1 Glacier surface elevation changes

We considered two contrasting patterns of glacier surface elevation change to be diagnostic of surge-type glaciers. We aimed to identify glaciers which exhibited substantial and widespread surface elevation gain (thickening) over their receiving zone, which is also commonly accompanied by surface elevation decrease (thinning) over a glacier's reservoir zone due to the flux of ice between the two areas (Fig. 1a). We consider this pattern of elevation change to be diagnostic of the active phase of a glacier's surge cycle. We also aimed to identify glaciers which exhibited substantial and widespread reservoir zone thickening and concomitant terminal thinning, which we attribute to quiescent phase ice build-up occurring alongside post-surge phase ice loss at lower elevations (Fig. 1b). To identify the mass displacement associated with the active phase of a glacier's surge cycle or substantial ice mass build-up associated with the quiescent phase of the surge cycle, we examined multi-temporal datasets of surface elevation change (d$H$) over HMA glaciers. We primarily used the d$H$ data generated by Shean et al. (2020), which cover the period 2000–2018, to identify the elevation change patterns described above. To aid the identification of surge-like behaviour which may have occurred at the beginning of the period covered by Shean et al. (2020), the signal of which may have been obscured by subsequent long-term thinning, we also examined d$H$ data generated by Hugonnet et al. (2021) over the periods 2000–2004 and 2005–2009 and by Brun et al. (2017) over the period 2000–2016 CE2. Distinctive surface elevation patterns were identified manually and corroborated by multiple users.

### 2.2 Glacier surface velocity

The NASA MEaSUREs Inter-mission Time Series of Land Ice Velocity and Elevation (ITS_LIVE) project (Gardner et al., 2019) provides measurements of glacier surface velocity at monthly to yearly temporal resolution over all major

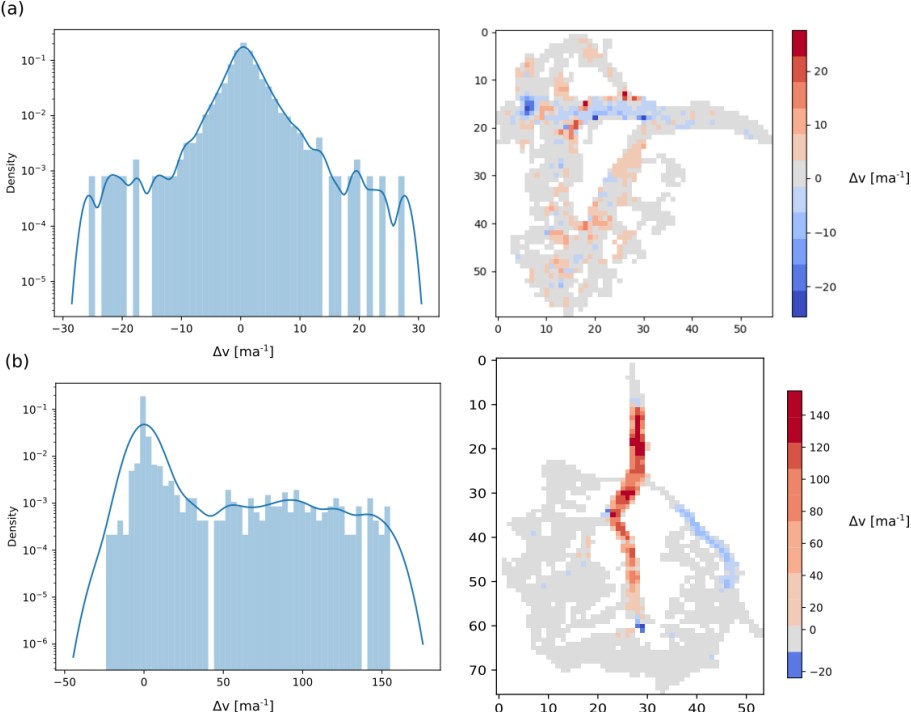

**Figure 2.** Distribution and map of d$V$ for two glaciers. **(a)** Non-surge-type glacier: Samudra Tapu Glacier in 2013, Chandra River basin, Western Himalaya. Note the symmetrical d$V$ distribution. **(b)** Surge-type glacier: K2 North Glacier complex in 2015, Karakoram. Note the positive heavy tail on d$V$ as a consequence of surging behaviour.

land ice regions, with a resolution of 240 m. Glacier surface velocities were estimated over the 1985–2018 period from Landsat 4, 5, 7 and 8 images using the auto-RIFT feature tracking algorithm (Gardner et al., 2018). As a consequence of unequal data quality and scarcity in the years covered by the earlier Landsat data archives, we only considered yearly glacier surface velocity derived from Landsat 7 and 8 imagery, between 2000 and 2018, which also matches the period covered by surface elevation change datasets.

The ITS_LIVE yearly glacier surface velocity data are provided with an associated error map. From the collection of yearly glacier surface velocity and their respective error maps, we then form $V_0$, the error-weighted mean velocity map for the study period, as follows:

$$V_0 = \frac{\sum_{i=1}^{n} w_i V_i}{\sum_{i=1}^{n} w_i}, \tag{1}$$

where $V_i$ is the glacier surface velocity dataset for year $i$, and the weights $w_i$ are defined as

$$w_i = \frac{1}{\epsilon_i^2}, \tag{2}$$

with $\epsilon_i$ being the error map for year $i$.

The yearly map of velocity change (d$V$) is computed as the difference between $V_0$ and $V_i$ for each year $i$. We then follow the method from Mouginot and Rignot (2015) and eliminate potential outliers using a $3 \times 3$ median filter.

From these data, we aim to identify positive velocity anomaly with a magnitude commensurate to that of a glacier surge. We thus assume the distribution of d$V$ to be a zero-mean Gaussian distribution for glaciers with stable flow over the studied period (Fig. 2a). Surging behaviour typically displays more complex patterns, and one can expect the distribution of surface velocity variations to be either positively (during the active phase) or negatively (quiescent phase) heavy-tailed (Fig. 2b). In the present work, we aim to identify glaciers displaying strong positive heavy tails resulting from active surges on particular years. We thus compute the range between the median ($P_{50}$) and the percentile 95 ($P_{95}$) for each glacier and year:

$$\mathrm{IPR}_i = |P_{95i} - P_{50i}|, \tag{3}$$

where $\mathrm{IPR}_i$ is the inter-percentile range for year $i$. From Eq. (3), we propose to quantify surge magnitude as a surge index, defined as follows:

$$s_i = \frac{\mathrm{IPR}_i}{k \cdot V_0}, \tag{4}$$

where $s_i$ is the surge index for year $i$. $k$ corresponds to a threshold for surge identification. Surge velocities are usually described as 10 to 1000 times greater than standard glacier velocity (Cuffey and Paterson, 2010; Jiskoot, 2011). However, surges in HMA have been documented with surface velocities close to 4 times the standard velocity (Quincey et al.,

2011, 2015; Bhambri et al., 2017; King et al., 2021). Thus, we here work with $k = 4$.

In practice, a glacier is defined as surge type when it presents at least one occurrence of $s_i >= 1$ within the whole study period. Equation (4) allows us to quantify the relationship between the observed velocity anomaly for each specific year and $V_0$. We thus highlight variations in the magnitude of the active phase of surge events from one year to another.

### 2.3 Glacier surface features

Substantial changes occur in the stress and strain regime at the surface of a surge-type glacier during its active phase which result in the widespread modification of surface structures such as crevasse patterns (Jennings and Hambrey, 2021). Such changes in structural glaciology have previously been used to identify glacier surging (see Grant et al., 2009; Lovell et al., 2018, for example).

We performed systematic visual checks for the presence and temporal variation of diagnostic crevasse patterns on very-high-resolution (VHR, spatial resolution finer than 5 m) optical imagery. We used multi-temporal, VHR imagery in Google Earth (GE) and additional images in Bing Maps to confirm or rebut the identification of surge-type glaciers, alongside glacier surface elevation and velocity changes. We aimed to identify crevasse patterns indicative of longitudinal extension or lateral shear margins, which are indicative of ice flow increases and ice mass redistribution in the down-glacier direction (Fig. 3), as well as potholes on the surface of the glacier which suggest ice flow stagnation. The availability of VHR optical imagery allowed for the examination of structures in glacier reservoir and receiving zones, whereas moderate-resolution optical imagery (Landsat, for example) typically only allows for the identification of larger glacier features such as looped moraines, which might not be prevalent on all surge-type glaciers

### 2.4 Glacier outlines and tributary glacier surges

In this study, we used the Randolph Glacier Inventory (RGI) V6.0 glacier outlines (Pfeffer et al., 2014; RGI Consortium, 2017). The RGI provides a snapshot of glacier extent near the beginning of the 21st century, and so current glacier extent, particularly of glaciers which have since surged, may differ somewhat from the area estimates associated with the RGI. However, such differences are unlikely to cause erroneous comparisons at a regional level, and we have not further modified RGI glacier extents in order to compare the attributes of surge-type glacier clusters in different parts of HMA.

Similarly, the RGI does not propose individualized polygons for tributary glaciers within larger glacier complexes. Any statistical analysis specifically studying spatial quantities of individual glaciers (surface area for example) from an RGI-based surge-type glacier inventory can therefore be biased by the presence of tributary glacier surges in wider, non-surge-type glacier complexes. Prominent examples of this phenomenon are Bivachny Glacier (Fedchenko Glacier complex, Pamir, Tajikistan), Maedan Glacier (Panmah Glacier, Karakoram, Pakistan) and Liligo Glacier (Baltoro Glacier, Karakoram, Pakistan) (Wendt et al., 2017; Goerlich et al., 2020; Hewitt, 2007; Belo et al., 2008).

In the present study, we manually digitized the maximal surface area impacted by a given surge within glacier complexes observed between 2000 and 2018. Surge-affected areas are delineated from the extent of substantial glacier surface elevation change caused by ice mass transfer during the active phase of a surge, using very-high-resolution optical imagery from Google Earth and Bing Maps to corroborate each delineation. We therefore provide an estimate of the maximum surface area affected by surges in glacier complexes ($n = 300/666$), in addition to the positively biased surface area of surge-type RGI polygons. Due to the relatively low resolution (size of 1 pixel on the ground) of the velocity data used in this study (240 m; see Sect. 2.2), we follow the method proposed by Dehecq et al. (2019) and only consider RGI polygons with an area greater than $5 \, km^2$ for velocity-based analyses.

### 2.5 Identification of surge-type glaciers

We carried out a preliminary manual search of surface elevation change datasets to highlight possible surge-type glaciers by identifying typical surge-induced surface elevation change patterns (see Sect. 2.1) which contrasted directly with local thinning patterns evident on non-surge-type glaciers. In parallel, automatic surge detection was carried out from the ITS_LIVE datasets on RGI polygons with an area greater than $5 \, km^2$. Two sub-inventories were thus created before being merged together. Each glacier was then individually inspected for the presence of surface features typical of active surging (see Sect. 2.3). To be qualified as surging, individual glaciers must display at least two of the three proposed identification criteria of rapid changes in surface elevation, surface velocity, surface crevassing and potholes. Geomorphological evidence of surging such as looped moraines, push moraines or ice strandlines is not considered as identification criteria. By combining different criteria, we typically mitigate false identification of phenomena other than surges which may cause a similar signal in, for example, surface velocity data such as lake-terminating glacier frontal speed-up. The multi-factor approach also mitigates problems such as remnant noise in the surface elevation and velocity datasets among others.

## 3 Results

Out of the $\approx 95\,000$ glaciers in HMA, 666 have displayed surge-type behaviour between 2000 and 2018 (Fig. 4). Traditional surge-type glacier clusters are clearly defined, as ap-

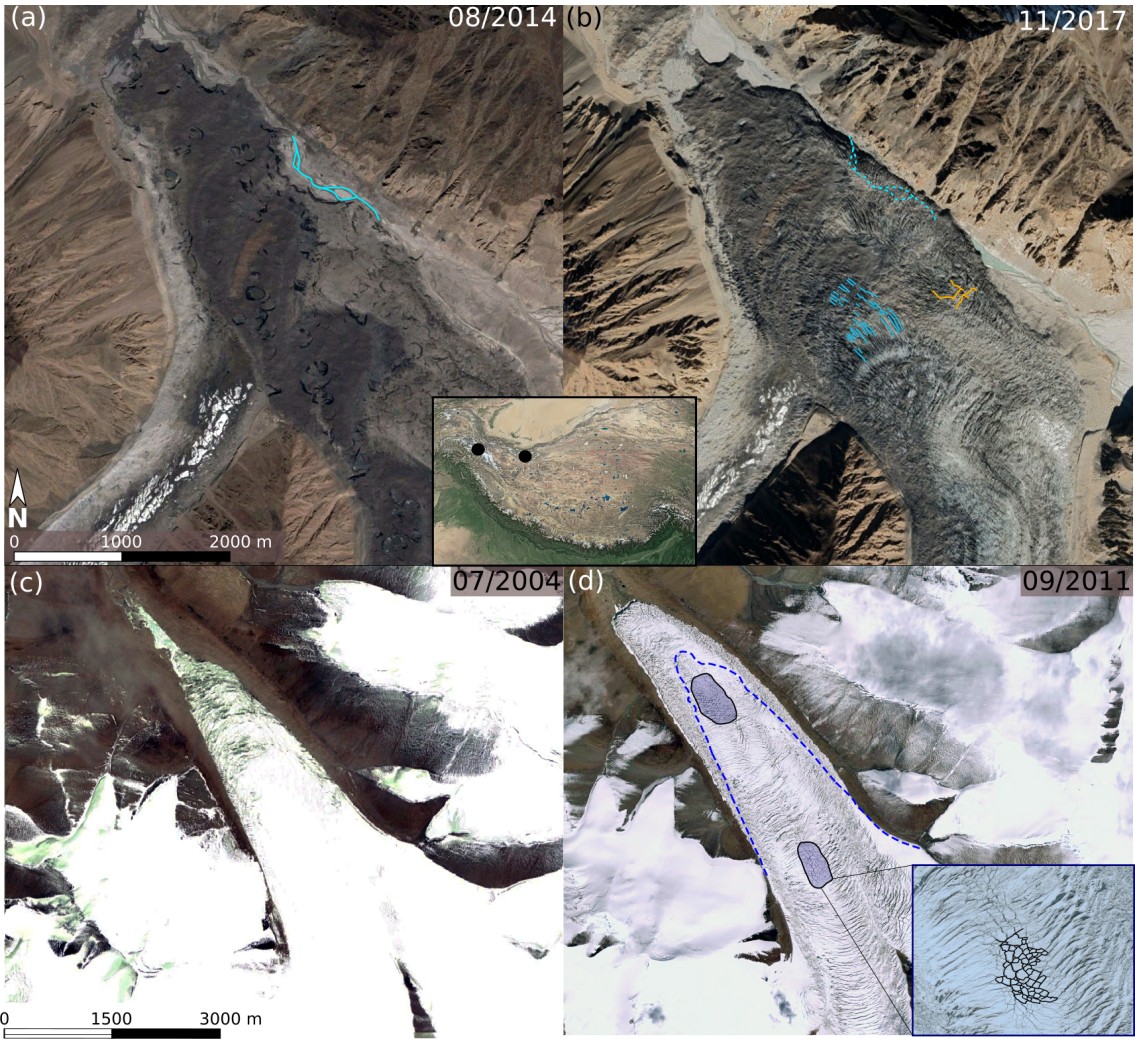

**Figure 3.** Google Earth imagery close-ups of surface features indicative of surges on debris-covered (**a, b**: 2017 surge of Khurdopin Glacier, Hunza Basin, Karakoram) and clean ice glaciers (**c, d**: 2008–2014 surge of N2 Glacier, West Kunlun Tibet, light blue CE3). (**a**) Khurdopin Glacier in August 2014 during the quiescent phase. Note the abundance of supraglacial ponds on the stagnant ice. Blue line: river bed. (**b**) Khurdopin Glacier in November 2017 during the active phase (peaking in May 2017) (Steiner et al., 2018). The surface is heavily crevassed with longitudinal (blue lines) and intersecting crevasses (yellow). The blue dashed line represents the location of the river bed in panel (**a**). (**c**) N2 glacier in a quiescent phase in 2004. (**d**) N2 Glacier in 2011 after active-phase peak (August 2009–November 2010) (Yasuda and Furuya, 2015). The blue dashed line represents the position of the glacier terminus in 2004. Note the substantial terminus advance and the two zones (blue polygons) of rhombic crevasse patterns, indicative of surge-type behaviour (Herzfeld and Zahner, 2001; Herzfeld et al., 2004). © Google Earth 2021 and © Microsoft.

proximately two-thirds of the identified surge-type glaciers are located in the Karakoram (223) or in the Pamirs (223) (Table 1). Smaller clusters are located in the Tien Shan and the Kunlun Shan, whilst isolated and less numerous examples are found across the Tibetan Plateau and its peripheral mountain ranges (Table 1). Among these smaller clusters, the Tien Shan and Kunlun Shan present the highest number of surging glaciers (74 and 73 respectively).

Our inventory is based on the fusion of two sub-inventories computed from the study of glacier surface elevation change (see Sect. 2.1) and surface velocity (see Sect. 2.2) datasets.

Analysis of glacier surface elevation changes revealed surge-like signals for 607 glaciers, while the sub-inventory automatically generated from ITS_LIVE was composed of 860 glaciers. Around 300 glaciers were common to both inventories. The substantial difference in the number of glaciers identified in the two sub-inventories can be explained by two main causes. First, the overall uncertainty and data voids in the ITS_LIVE datasets will lead to artificial velocity anomalies from one year to another. Second, in line with the results of Pronk et al. (2021), we have noted the presence of a large number (at least 70, exact number not known) of lake-

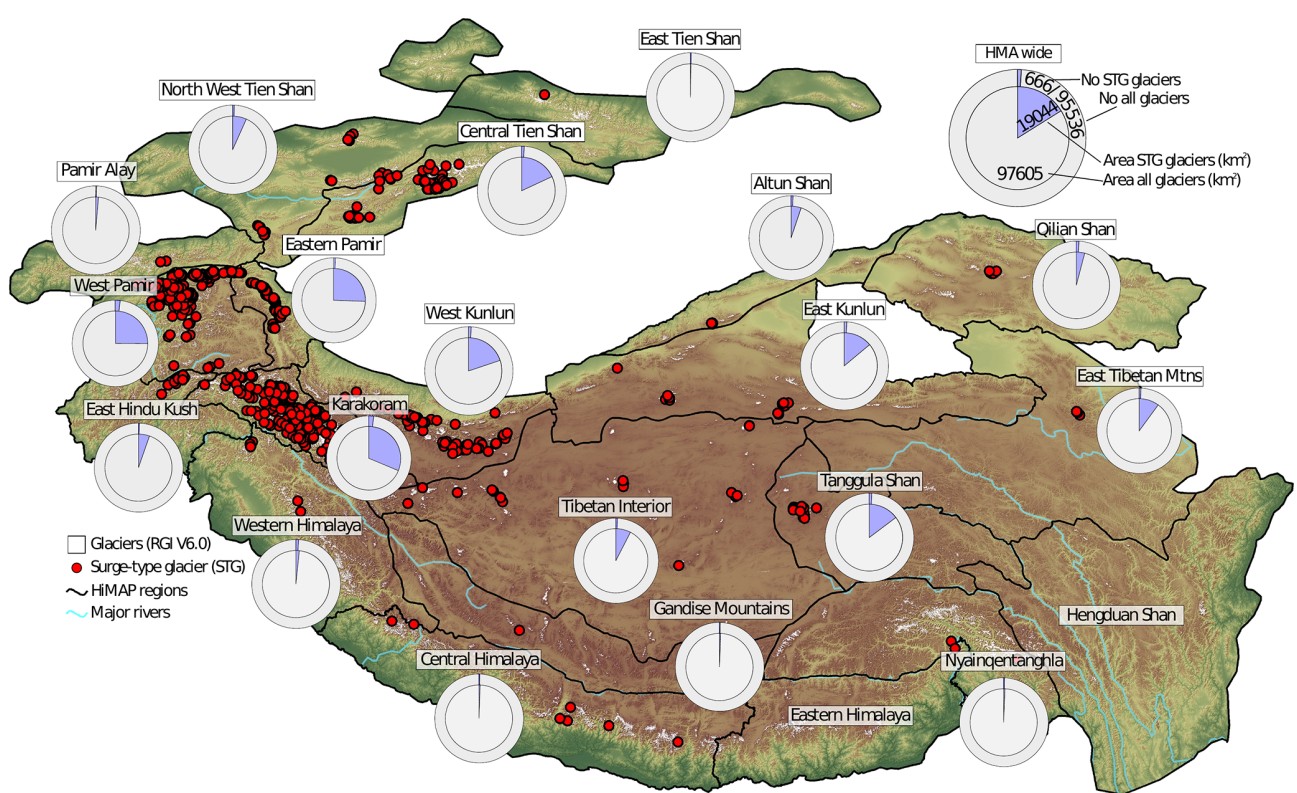

**Figure 4.** The distribution of surge-type glaciers (red dots) across HMA, identified by our multi-factor remote sensing approach. The number and area of surge-type glaciers within each HIMAP region (Bolch et al., 2019), compared to the total glacier population, are also shown. Prominent but well-known clusters of surge-type glaciers are evident in the Pamirs and Karakoram, but our analysis has also confirmed and detected smaller clusters of surge-type glaciers across the Tibetan Plateau and Tien Shan CE4.

terminating glaciers displaying contrasted surface velocity signals.

From the 666 identified glaciers, 68 had been previously reported as "observed surging" in the RGI V6.0. We further confirm 107 glaciers, previously identified as "probable" or "possible" surge-type glaciers. Finally, we newly identify 491 glaciers, previously categorized as "no evidence" and "not assigned". Compared to the Sevestre and Benn (2015) inventory, we newly identified 482 surging glaciers: 140 in the Pamirs (see Sect. 4.2 for more), 150 in the Karakoram and 63 in the Tien Shan. Sevestre and Benn (2015) did not identify surges in the Kunlun Shan, Himalayas (King et al., 2020), Tibetan Mountains or Qilian Shan.

RGI polygons of surge-type glaciers cover a total surface area of 19 044 km$^2$, which corresponds to $\approx 19.5\%$ of the glacierized area in HMA. Unsurprisingly, surge-type glaciers account for the greatest portion of RGI glacier area in the Karakoram (45.3 %), followed by the Eastern (33.9 %) and Western Pamir (35.6 %) (Fig. 4 and Table 1). Surge-type glaciers account for less than 1 % of glacier area in 9 of the 22 HIMAP (Bolch et al., 2019) regions. We find no evidence of glacier surging in the Hengduan Shan or Dzungarian Alatau.

## 3.1 Distribution and geometry of surge and non-surge-type glaciers

Previous studies have demonstrated significant differences in glacier geometry (e.g. length, slope) between surge-type and non-surge-type glaciers (see Sevestre and Benn, 2015, among others, for example). We here further document the impact of geometry on surge behaviour by studying glacier elevation range, slope, length area and aspect, using attributes derived from the RGI V6.0. Note that, due to the wide ranges covered by each attribute, we compare common logarithm values.

Throughout HMA, surge-type glaciers systematically present greater elevation range and area (Fig. 5). As glacier length, slope and elevation range are strongly correlated attributes, such a pattern is unsurprising. We further observe generally shallower slopes for surge-type glaciers than non-surge-type glaciers. The difference is not as clear cut as for the other parameters; the median surge-type glacier slope indeed lies within CE5 1 standard deviation of the median for non-surge-type glaciers. Slope alone is thus not a sufficient predictor for surge-type behaviour in HMA. We however observe that the frequency of surge-type glaciers drastically in-

**https://doi.org/10.5194/tc-16-1-2022** **The Cryosphere, 16, 1–22, 2022**

**Table 1.** Table of number and proportion of area covered by RGI polygons of surge-type glaciers for each HIMAP region (Bolch et al., 2019). STG: surge-type glacier.

| HIMAP region | Greater HIMAP region | $N$ STGs | $N$ glaciers | Area of STGs (km$^2$) | Area of all glaciers (km$^2$) | STG area (%) |
|---|---|---|---|---|---|---|
| Karakoram | Karakoram | 223 | 11 586 | 9736 | 21 475 | 45.3 |
| Western Pamir | Pamirs | 173 | 9118 | 2883 | 8480 | 33.9 |
| Eastern Pamir | Pamirs | 47 | 1609 | 784 | 2200 | 35.6 |
| Pamir Alay | Pamirs | 2 | 3151 | 26 | 1846 | 1.4 |
| Central Tien Shan | Tien Shan | 54 | 5836 | 1629 | 7270 | 22.4 |
| Northern/Western Tien Shan | Tien Shan | 19 | 3903 | 173 | 2261 | 7.7 |
| Eastern Tien Shan | Tien Shan | 1 | 4259 | 3 | 2333 | 0.1 |
| Western Kunlun Shan | Kunlun Shan | 60 | 5674 | 2096 | 8456 | 24.8 |
| Eastern Kunlun Shan | Kunlun Shan | 13 | 3092 | 485 | 2994 | 16.2 |
| Tibetan Interior Mountains | Tibetan Mountains | 19 | 3493 | 372 | 3815 | 9.8 |
| Tanggula Shan | Tibetan Mountains | 14 | 1584 | 318 | 1840 | 17.3 |
| Eastern Hindu Kush | Eastern Hindu Kush | 9 | 4375 | 193 | 3051 | 6.3 |
| Qilian Shan | Qilian Shan | 9 | 2684 | 68 | 1597 | 4.3 |
| Central Himalaya | Himalayas | 7 | 7374 | 55 | 8986 | 0.6 |
| Western Himalaya | Himalayas | 6 | 9951 | 136 | 8117 | 1.7 |
| Eastern Himalaya | Himalayas | 1 | 2963 | 9 | 2983 | 0.30 |
| Altun Shan | Kunlun Shan | 3 | 467 | 16 | 295 | 5.4 |
| Eastern Tibetan Mountains | Tibetan Mountains | 3 | 522 | 35 | 312 | 11.2 |
| Nyainqêntanglha | Tibetan Mountains | 2 | 7417 | 16 | 7046 | 0.2 |
| Gangdise Mountains | Tibetan Mountains | 1 | 3851 | 11 | 1270 | 0.9 |
| Hengduan Shan | Hengduan Shan | 0 | 2056 | 0 | 1281 | 0 |
| Dzungarian Alatau | Dzungarian Alatau | 0 | 968 | 0 | 520 | 0 |
| Total/mean* | | 666 | 95 536 | 19 044 | 97 605 | 19.5* |

* Refers to the fact 19.5 % in the last column is a mean, while all other columns are totals.

creases with greater lengths and shallower slopes (Fig. 6), as predicted by the enthalpy balance theory (Benn et al., 2019).

While in the Tien Shan, Pamirs and Kunlun Shan, aspect distributions do not significantly differ, we nonetheless note narrower distributions and increased number in south quadrants for surge-type glaciers, with $\approx 8\%$ of glaciers flowing S in the Tien Shan, $\approx 7\%$ flowing SW and S in the Pamirs, and $\approx 9\%$ flowing SE in the Kunlun Shan ($\approx 2\%$ for non-surge-type glaciers in all three regions) (Fig. 7). We note distinct variability in the aspect distribution of surge-type glaciers in the Tibetan Mountains (Fig. 7), which show no dominant orientation. Inter-regional differences in glacier type (defined in the RGI V6.0) reflect and possibly explain the aforementioned variability in glacier aspect. Topographically confined mountain glaciers are most common in regions with coherent surge-type glacier aspect (Fig. 8), whereas regions with larger numbers of ice cap outlet glaciers show more variability in surge-type glacier aspect (Fig. 7). Indeed, the number of outlet surge-type glaciers outweighs the number of surge-type mountain glaciers in the Tibetan Mountains, where there is no clear aspect preference of surge-type glaciers.

## 3.2  Analysis of glacier geometry and its impact on surge-type behaviour

We here further document the potential impacts of geometry on surging behaviour, both at HMA and regional scales. More specifically, we study the relationships between surge-affected area in glacier complexes and median surge indices with glacier median elevation, elevation range, slope and length. We observe no particular relationship between the area affected by surge in glacier complexes and the median elevation or the slope (Fig. 9). Similarly no evident correlation can be described between the median surge index and the same geometric attributes. The common logarithm of the surge-affected area in glacier complexes is however strongly linearly correlated (Pearson's $R = 0.81$, $p$ value $< 0.001$, zero-mean residuals) to that of maximum glacier complex length. This linear correlation of the common logarithm of two variables yields a power law relationship between the surge-affected area and the maximum length of glacier complexes, with a power law exponent of 1.3. We further identify similar linear relationships between the common logarithms of the median surge index and the elevation range (Fig. 9) (Pearson's $R = 0.59$, $p$ value $< 0.001$, zero-mean residuals), with a power law exponent close to 0.84.

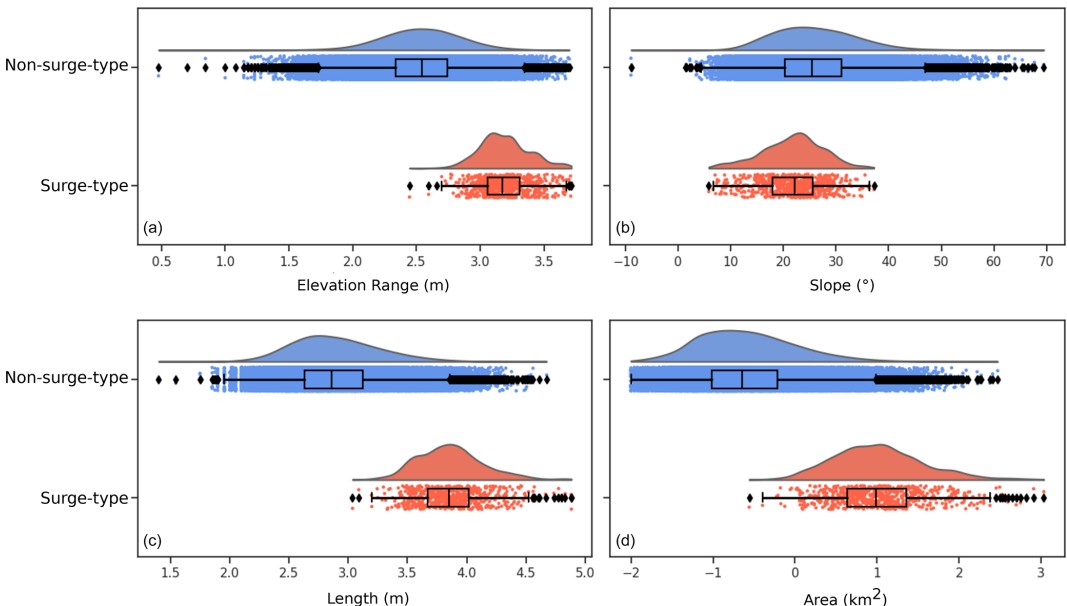

**Figure 5.** Distributions of geometrical attributes between non-surging and surge-type glaciers: **(a)** elevation range, **(b)** slope, **(c)** length and **(d)** area. Surge-type glaciers are represented in orange and non-surge-type glaciers in blue. $L_{\max}$ is the length of the centreline of the longest glacier trunk of each RGI polygon. Area, length and altitude range are represented using common logarithms. Whiskers represent the 95th percentile of each distribution, while limits of the box represent 1 standard deviation.

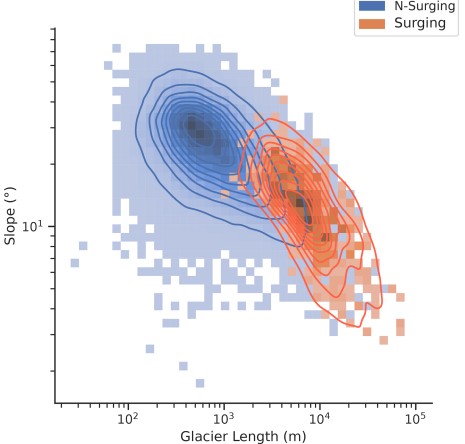

**Figure 6.** Length and mean slope for surging and non-surge-type glaciers in HMA. Due to the disparity in sample sizes between surge-type (666) and non-surge-type glaciers (more than 90 000), the length–mean-slope relationship is represented using a bi-variate histogram. Contour lines represent an estimate of the probability density, approximated using kernel density estimation.

On a regional scale, the Tien Shan, Pamirs and the Karakoram all show relatively high correlation between range and median surge index with statistically significant $p$ values. We note an important anti-correlation between slope and surge area in the Kunlun, which is a consequence of contrasting glacier geometries in the Kunluns. While the distribution of glacier slopes in the Karakoram and the Pamirs (for example)

presents a unimodal distribution with respective means close to 25 and 22°, slope distribution in the Kunlun is uniform with no clearly identifiable mode. All identified correlations display a heavy scatter, and more data are needed to consolidate these relationships.

### 3.3 The impact of surge-type behaviour on glacier mass balance

The glacier-wide mass balance measurements presented here originate from the studies of Shean et al. (2020) and Hugonnet et al. (2021), covering the periods 2000–2018 and 2000–2020, respectively.

We observe no major difference between the medians of the different distributions in the Karakoram, Pamirs, Kunlun Shan, Tien Shan and Tibetan Mountains (Fig. 13). Median mass budgets for surge-type glaciers are close to $-0.2\,\text{m.w.e.}\,\text{a}^{-1}$ in the Karakoram, $-0.1\,\text{m.w.e.}\,\text{a}^{-1}$ in the Pamirs and $-0.09\,\text{m.w.e.}\,\text{a}^{-1}$ in the Kunlun Shan. Non-surging-glaciers in the same regions display median mass losses close to $-0.09$, $-0.05$ and $-0.05\,\text{m.w.e.}\,\text{a}^{-1}$ respectively. We however note that the mass balance distribution for surge-type glaciers in the Karakoram is positively skewed with $g = 1.1$ and negatively skewed with $g = -1.4$ for non-surge-type glaciers. The density of the distribution for surge-type glaciers in the Karakoram is thus concentrated towards more negative mass balance. Due to the relatively low number of surge-type glaciers reported in the Qilian Shan, Eastern Hindu Kush and Himalayas, we do not consider results for these regions as representative.

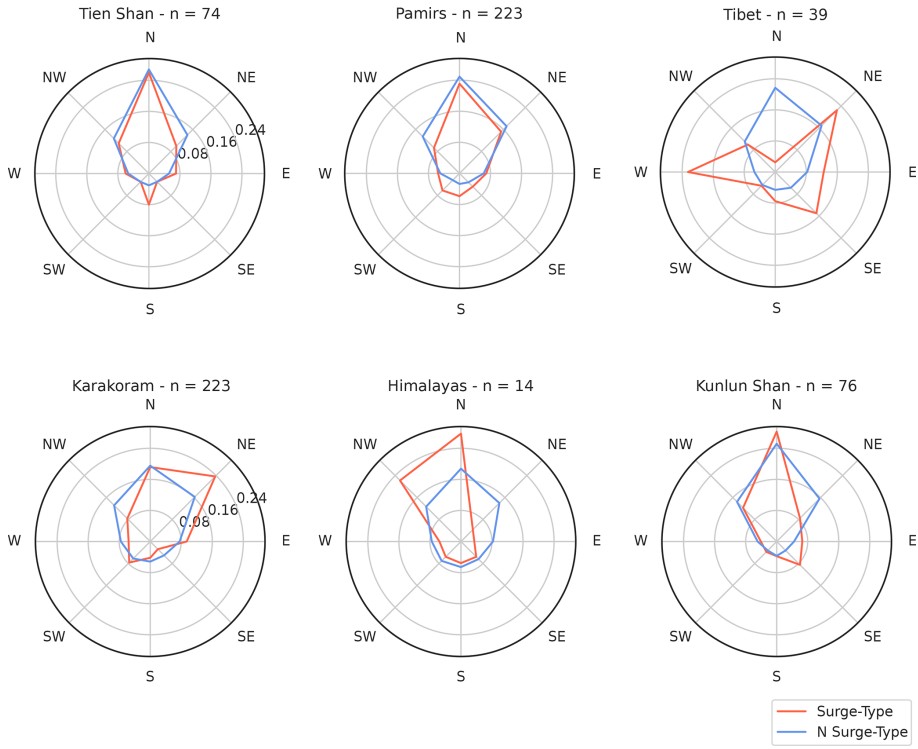

**Figure 7.** Aspect sector distribution for surging and non-surge-type glaciers for each greater study region per count. *n* refers to the number of surge-type glaciers per greater HIMAP region.

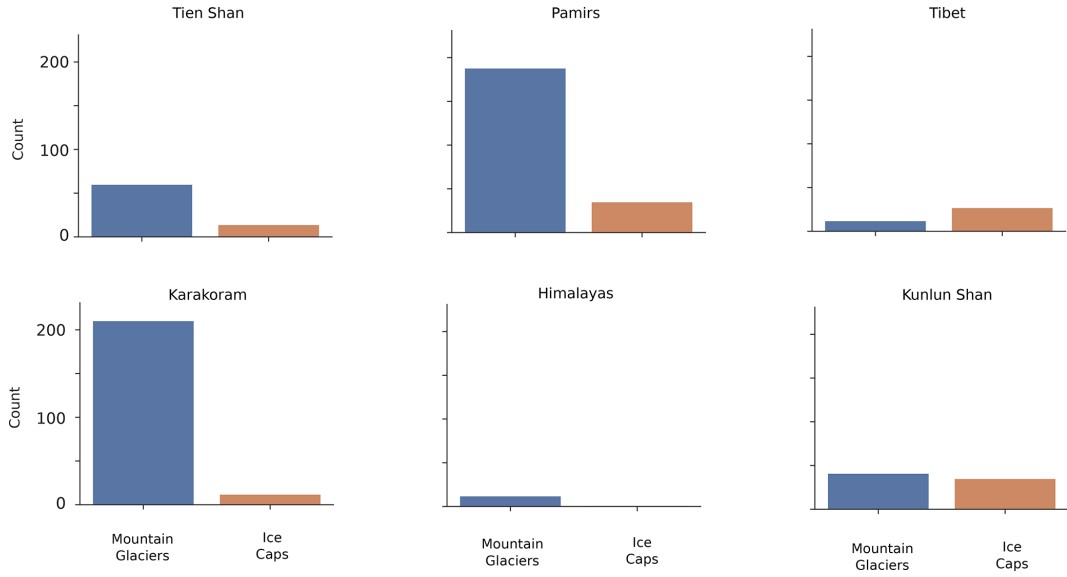

**Figure 8.** Distributions of surge-type glacier types for each of the greater HIMAP regions. Note that the Tibetan Mountains are the only region where ice cap outlet glaciers are more represented than mountain glaciers, with a ratio of 2.7.

Inspection of the temporally resolved mass balance estimates of Hugonnet et al. (2021) suggests marked changes of glacier mass budgets during their transition from late quiescence to their active surge phase and then immediately following their latest surge (Table 2). More specifically, Khurdopin Glacier displayed balanced and even positive mass balance prior to (2010–2015) and during the early stages (2015–2016) of its most recent surge. Khurdopin Glacier then experienced substantial ice mass loss ($-0.22$ to $-0.3$ m w.e. a$^{-1}$) during the later stages of its latest surge and immediately fol-

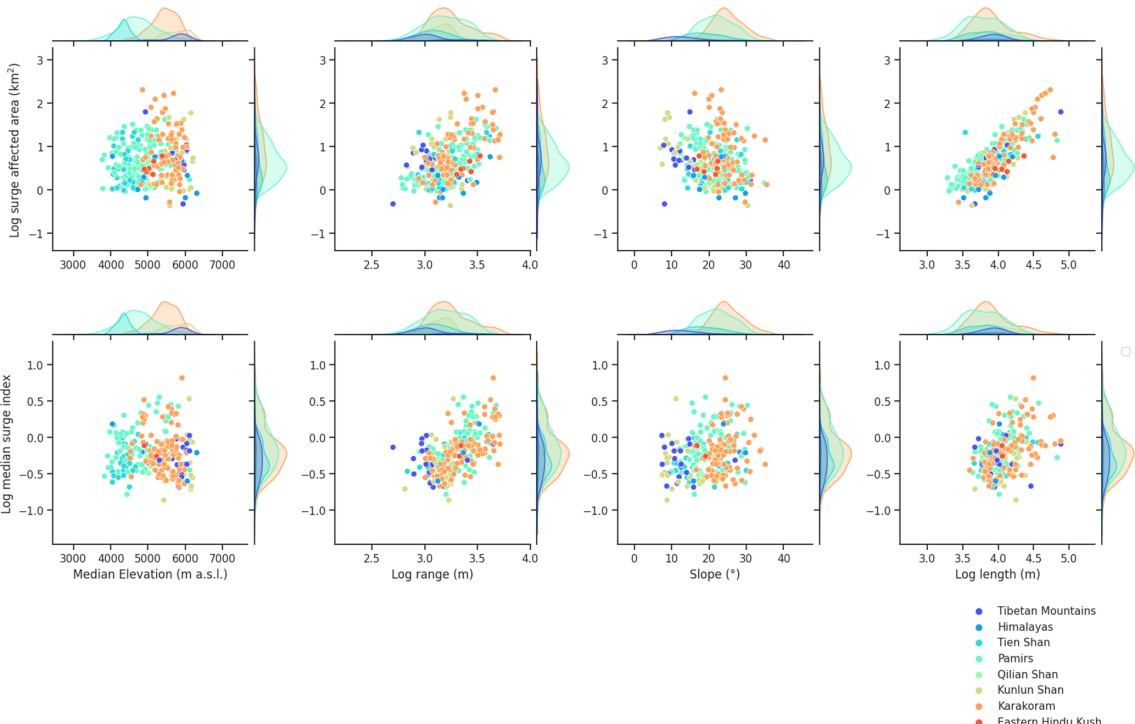

**Figure 9.** Correlation between the different geometric parameters of surge-type glaciers.

lowing surge cessation. Similarly, Hispar Glacier (not including the Kunyang tributary, which itself surged around 2007) experienced a slight decrease in mass loss between the later stages of quiescence (2010–2012) and the start of its surge phase (2012–2014 – Table 2) and then a significant increase in its rate of ice loss once the active phase of the surge had ended. The rate of ice mass loss doubled from Hispar Glacier immediately following the end of its 2014–2016 surge.

### 3.4 Temporal variability and maximum velocities in surge active phase

We observe no major differences in the length of the active phase of surges captured by the velocity data in different regions across HMA, the distributions of which are presented in Fig. 10a and b respectively. While the distributions present similar medians (2, 2, 2.3 and 2.3 years for the Tien Shan, Pamirs, Kunlun Shan and Karakoram respectively) and 25th percentile (1.2, 1.3, 1.3 and 1.4 years), we note a disparity in 75th percentile (2.2, 3.4, 4.4 and 4 years). For the Pamirs, Kunlun and Karakoram, the inter-quartile range exceeds the median maximum surge index by a factor 1.1–1.3, highlighting the irregularity of surge velocities. Both regions notably display surge indices greater than 10, with maxima up to 18 in the Kunlun Shan and 223 in the Karakoram. Velocity data for Qilian Shan, Eastern Hindu Kush, Himalayas and Tibetan Mountains were too sparse to be considered representative (< 10 samples in each case).

We note no clear difference in surge active-phase duration between regions (Fig. 10c). The majority of surges typically last less than 3 years, with a median duration of $2.6 \pm 0.1$ years. However, the distributions are systematically heavy-tailed, with all regions displaying active phases lasting 18 years.

We finally studied the distributions of duration and cumulative sum of IPR (see Eq. 3) for each surge over HMA. We used the latter as a crude proxy of the total energy dissipated during a given surge.

Both distributions are presented in Fig. 11 and display the characteristic shape of power-law-like (or Pareto-like) distributions. Power laws are common in geophysical systems where energy accumulated over a significant period is released rapidly (also known as avalanching systems) and arise from a wide variety of physical phenomena (criticality and self-organized criticality (SOC), among others; see e.g. Turcotte, 1989; Sachs et al., 2012; Åström et al., 2014, and Corral and González, 2019, for more). To further study the hypothesis that the velocity anomaly during active phases and the duration of surges follow power law distributions, we use the widespread method of Clauset et al. (2009), relying on maximum likelihood estimation of power law distribution parameters (Pawitan, 2001; Bauke, 2007) and Kolmogorov–Smirnov distance minimization or log-likelihood ratio goodness-of-fit tests (see Park and Kim, 1992; Press et al., 2007, for example). The proposed power law model fits the velocity anomaly during active phases bet-

**Table 2.** Example of mass balance (m.w.e. a$^{-1}$) over a specified period and phases in the surge cycles of Khurdopin and Hispar glaciers' last documented surges. Data from Hugonnet et al. (2021). Note the substantial uncertainties associated with each elevation change estimate. For more on the specific surges of each glacier, we refer the reader to Paul et al. (2017), Bhambri et al. (2020) and Steiner et al. (2018).

| Glacier | Quiescence | Build-up | Active phase | Post-surge |
|---|---|---|---|---|
| Khurdopin | $-0.04 \pm 0.3$ (2010–2015) | $0.17 \pm 1.4$ (2015–2016) | $-0.22 \pm 1.4$ (2016–2017) | $-0.3 \pm 1.6$ (2017–2018) |
| Hispar (main trunk) | $-0.38 \pm 0.6$ (2010–2012) | $-0.24 \pm 0.6$ (2012–2014) | $-0.4 \pm 0.6$ (2014–2016) | $-0.8 \pm 0.6$ (2016–2018) |

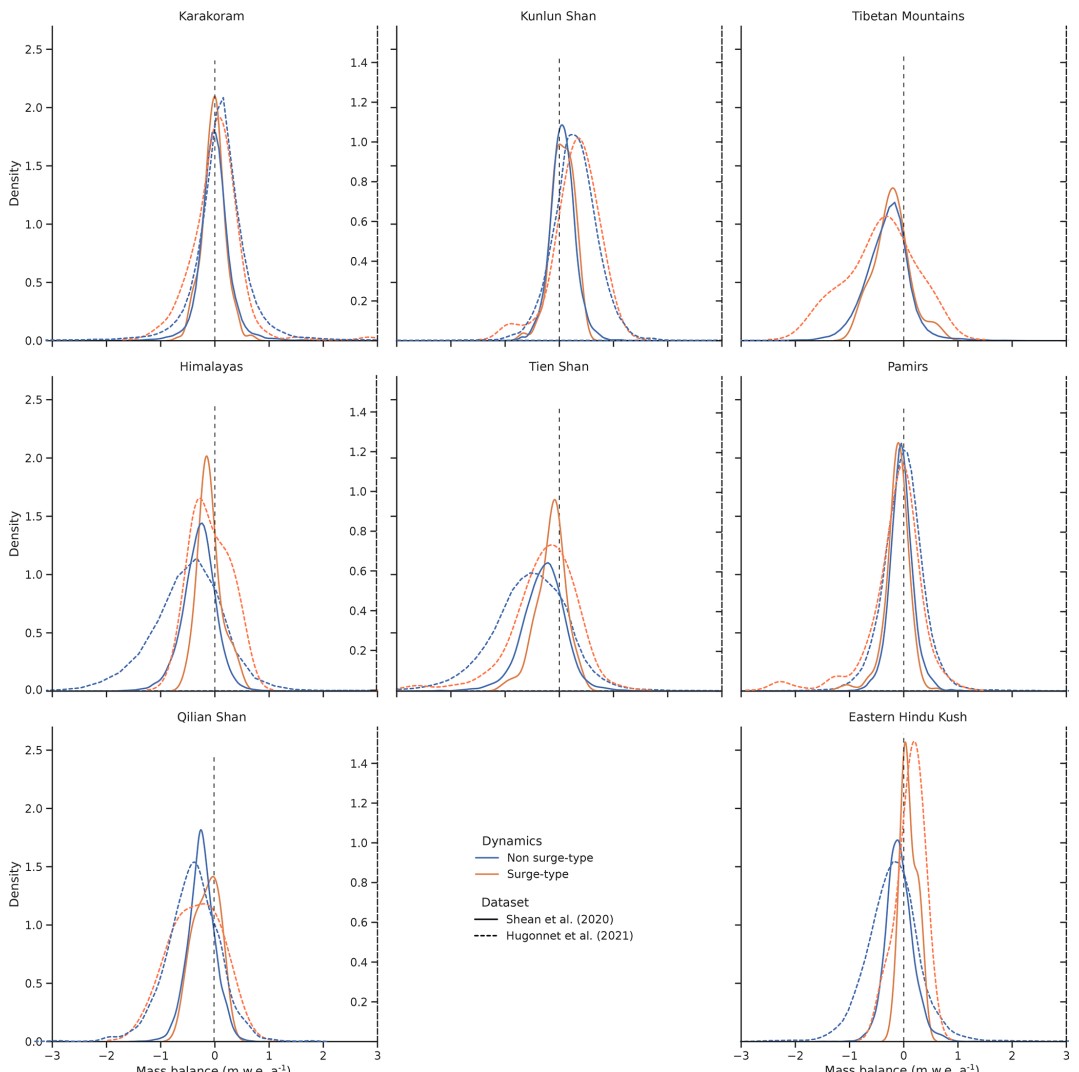

**Figure 10.** Distribution of **(a)** median surge index per glacier, **(b)** maximum surge index per glacier and **(c)** duration of the longest active phase per glacier for the whole HMA and four of the larger HIMAP regions. Dashed lines represent the median of each distribution; dotted lines are the 25th and 75th percentiles.

ter than exponential models, with log-likelihood ratio ($R$) significantly greater than zero (see Table 3). On the other hand, the exponential model fits the duration of active phases of surges better than the best-fit power law model, with a log-likelihood ratio of −6.

The fits between the estimated probability densities and the data are further presented in Fig. 12. We find that the theoretical complementary cumulative distribution function (CCDF) is in good agreement with the cumulative sum of IPR data. As mentioned above, the fitted distribution does not replicate the active-phase duration distribution as well. This is likely to be a consequence of our heavily truncated observation period: due to the length of our study period we do not highlight surges with an active phase greater than 18

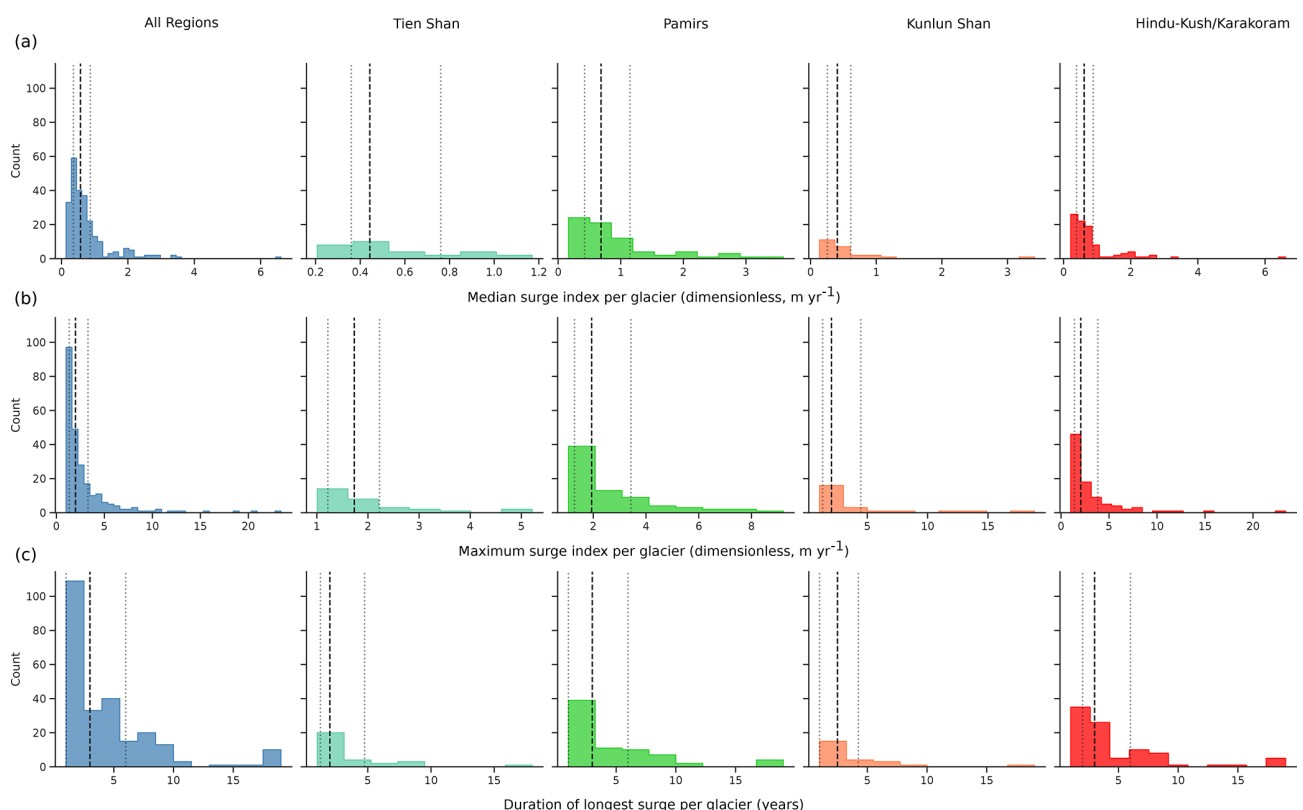

**Figure 11.** Distributions of surge duration and cumulative sum of IPR for the 439 surges recorded over the 245 glaciers presenting velocity data.

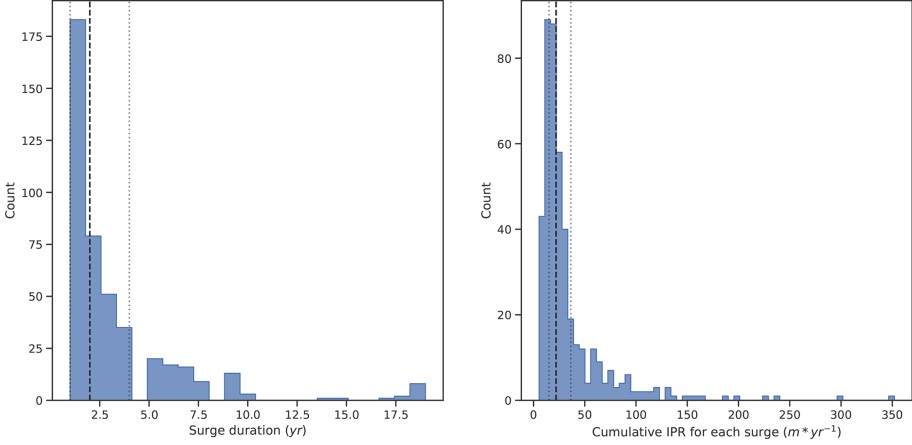

**Figure 12.** Estimation of empirical probability densities for the cumulative sum of IPR (blue) and surge active-phase duration (red). Solid black lines indicate power law fits.

years and thus truncate the power law's heavy tail. Similarly the yearly sample rate of the ITS_LIVE data used prevents identification of active phase lasting less than a year, further truncating the potential power law.

## 4 Discussion

### 4.1 Uncertainties

The surge-type glacier identification criteria which we have followed are built on a number of different publicly available datasets generated in previous studies. Each of these datasets is an imperfect representation of real geophysical signals and

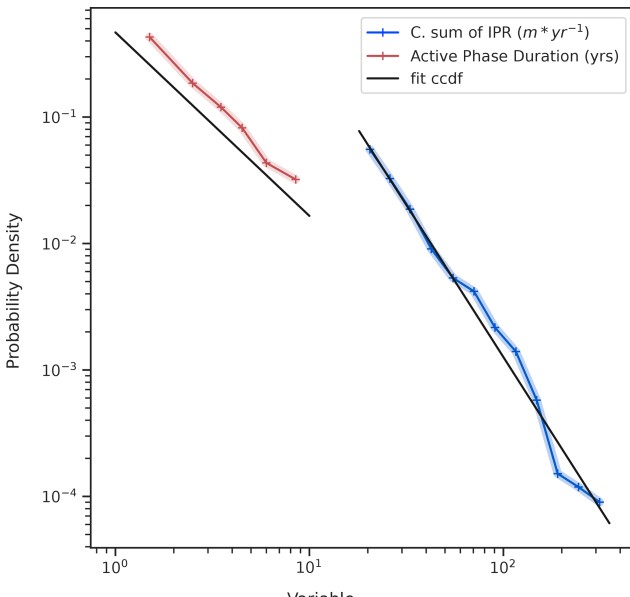

**Figure 13.** Distributions of mass balance for surge-type and non-surge-type glaciers for the greater HIMAP regions derived from the results of Shean et al. (2020) and Hugonnet et al. (2021). Strong differences are noticeable in regions where sample sizes are low (Himalayas, Qilian Shan, Hindu Kush, etc.). Comparisons between mass balance patterns of surge-type and non-surge-type glaciers are however difficult given the overall uncertainty associated with surface elevation change datasets.

**Table 3.** Table of power law parameters: $\alpha$ represents the power law exponent. $\sigma$ is the standard deviation associated with $\alpha$. $R$ represents the log-likelihood ratio between power law and exponential distributions.

| Variable | $\alpha$ | $\sigma$ | $R$ |
|---|---|---|---|
| Cumulative sum of IPR (m yr$^{-1}$) | 2.4 | 0.04 | 24 |
| Duration of surge active phase (years) | 1.45 | 0.03 | −6 |

attributes and presents a level of uncertainty. The uncertainty associated with the elevation change datasets of Hugonnet et al. (2021) varies between different time periods: between 2000–2004 and 2005–2009 the mean uncertainty over glacier surfaces in our inventory is 1.04 and 0.97 m a$^{-1}$, respectively. The mean uncertainty over the same glacier surfaces in our inventory using data from Brun et al. (2017) is 0.43 m a$^{-1}$, although we note lower coverage of this elevation change dataset over higher reaches of glaciers, where associated uncertainty would also be greater. Uncertainty estimates associated with the elevation change dataset of Shean et al. (2020) are not freely available, but we would expect similar levels of uncertainty to those of Hugonnet et al. (2021) to be associated with these data. Hugonnet et al. (2021) and Shean et al. (2020) estimate similar levels of uncertainty alongside their geodetic glacier mass balance estimates, and as surface

elevation change uncertainty tends to dominate the geodetic mass balance budget, the uncertainty associated with the underlying elevation change data should therefore be similar. Importantly, the uncertainty associated with elevation change datasets over glacier tongues lies predominantly well below the magnitude of the changes associated with surge behaviour (commonly 3–5 a$^{-1}$, occasionally as high as 10 a$^{-1}$ of thickening during intense surge events).

The uncertainty associated with the surface velocity data similarly varies between different periods, due to the switch from Landsat 7 to Landsat 8 imagery. Between 2000–2013 and 2013–2018 the median uncertainty of the ITS_LIVE dataset over glaciers in the inventory is around 1.35 and 0.37 m a$^{-1}$ respectively. The maximum uncertainty is significantly larger for 2000–2013 than 2013–2018, with 63 and 11.3 m a$^{-1}$. Uncertainties in the ITS_LIVE surface velocity dataset typically cluster in the accumulation area of glaciers, where trackable features are less abundant. The use of error-weighted surface velocity fields (Sect. 2.2) ensures a greater penalization of aberrant surface velocity pixels in the accumulation area. In the ablation area and over glacier tongues, uncertainties on surface velocity lie well below the magnitude of changes in velocity resulting from a surge (up to 150 m a$^{-1}$).

### 4.1.1 Identification of surge-type glaciers

Our study combines different, individual criteria used in previous studies to identify glacier surging, each of which may be more or less reliable in the detection of glacier surging in different scenarios. For example, the identification of glacier surging through the monitoring of glacier terminus positions is effective when glacier terminus advance is anomalous in comparison to the local pattern of glacier retreat in response to climate conditions, or where glaciers are devoid of stagnant, debris-covered tongues which may inhibit surge propagation towards glacier termini (Quincey et al., 2015). Glacier surging and mass-gain-related terminus advance may be confused in some scenarios (Lv et al., 2020), and additional observations are required to more accurately interpret glacier behaviour. In this study we qualify a glacier as surging if, and only if, it presented at least two of the three discriminatory criteria outlined in Sect. 2.1 to 2.3 (rapid surface elevation and/or velocity variations and/or intense surface crevassing and/or potholes). We are therefore confident that our interpretations are robust and that the prevalence of surging we document is an improvement on approaches using fewer surging criteria.

### 4.1.2 Identification of individual surges, temporal variability and IPR

In Sect. 3.4, we derived distributions of surge duration and intensity for every surge of each of the 245 glaciers with velocity data. While all glaciers identified have been system-

atically verified to be surging, individual surges of the same glacier are only derived from the velocity data. The determination of individual surges' onset and termination year therefore only relies on the velocity dataset; the duration of surges is thus known up to a certain level of certainty, depending on our threshold for identification ($k$ in Eq. 4). We tested the implications of different values of $k$ over the statistical attributes of the distributions of surge duration and cumulative sum of IPR and found that increasing $k$ naturally reduces the number of detected surges (data not shown). However, the distribution of surge durations does not differ substantially for different values of $k$. On the other hand, the distribution of cumulative sum of IPR is positively biased, as a result of the truncation of lower values. Increasing the detection threshold does not affect the shape of the distributions.

## 4.2 Comparison with other inventories

Inventories of surge-type glaciers have been assembled for many sub-regions of HMA, and particularly detailed studies have been carried out where hazards associated with surging have been widespread and recurring, such as the Karakoram and the Pamirs. A careful comparison of the similarities and differences between our inventory and those of previous studies is difficult because of the contrast in the methods used to identify glacier surging, the criteria used to define glacier surging and because the periods examined to identify glacier surging vary substantially between studies.

Sevestre and Benn (2015) proposed the only existing regional inventory of surge-type glaciers in HMA. The number of surge-type glaciers documented in our inventory significantly differs from that of Sevestre and Benn (2015), especially in the Pamirs. A further examination of the Sevestre and Benn (2015) inventory reveals that, out of the 827 surge-type glaciers documented, 284 correspond to individual tributaries within glacier complexes which are not individualized in the present study. From the remaining 543 glaciers, 35 documented in the RGI V5.0 (on which Sevestre and Benn, 2015, is based) do not exist in the RGI V6.0. Furthermore, we found that the proposed inventory and the one from Sevestre and Benn (2015) only share 83 identified surge-type glaciers in the Pamirs. This yields a difference of 390 surge-type glaciers between the two inventories. Upon further examination of the remaining glacier population from Sevestre and Benn (2015) we note a median glacier area of $1.6\,\mathrm{km^2}$, a sixth of the median area of the surge-type glacier population described in the present inventory ($9.6\,\mathrm{km^2}$). Of those 390 in the Sevestre and Benn (2015) inventory, 30 % present an area smaller than $1.0\,\mathrm{km^2}$. Close examination of these glaciers (Ujsu Glacier and the glaciers in its direct vicinity such as Aldzhaylau and Rakzou glaciers) using the surface elevation and surface velocity change data over the period 2000–2018 did not yield any evidence of surge-type behaviour. We rather observed constant glacier mass loss and recession, with no clear signal of instability-related velocity anomalies. Fur-

thermore, no mention of such a high number of surge-type glaciers in the Pamirs can be found in the literature used by Sevestre and Benn (2015). Kotlyakov et al. (2008) indeed refer to Osipova et al. (1998), mentioning "630 glaciers with indications of dynamic instability, 51 of them identified as surging"; they later state that 55 surge-type glaciers had been documented up to 2006 in the Pamirs. Given these unclear, conflicting and interpretive records of unstable glacier dynamics, the small sizes of glaciers in that population compared to other surge-type glaciers in the present and other studies (see next section), the lack of evidence for unstable behaviour over the 2000–2018 period, and the new considerations laid by the enthalpy balance theory (Benn et al., 2019), we here question the surge-type behaviour of this subset of 390 glaciers in the Pamirs. Assuming we were to consider that population of glaciers as indeed surge type, the estimated glacierized area covered by surge-type RGI polygons in HMA would become 20.6 %.

Further general comparisons can be drawn between inventories based on remotely sensed imagery to establish the success of the methods used however (Table 4). The inventory of Goerlich et al. (2020) is based on observations of surface elevation change and glacier terminus advance in the Pamirs over the period 1968–2018, which allowed for the identification of 206 surge-type glaciers. Our observations of elevation change, surface velocity change and structural glaciology between 2000–2018 suggest 223 surge-type glaciers in the same region. The limited difference between these results may suggest that few surges in the Pamirs exhibit low-magnitude elevation changes (e.g. Fig. 1) and require additional observations (surface velocity) to aid surge-type glacier identification. The low rate of thinning and ice mass loss in the Pamirs (see Shean et al., 2020, for example) has also likely preserved the strong signal of elevation change associated with surging in multi-decadal geodetic studies.

In contrast, Rankl et al. (2014) identified 101 surge-type glaciers in the Karakoram from glacier surface velocity and glacier terminus observations over the period 1972–2012, which is substantially less than the 223 identified in this study using data covering the period 2000–2018. The main methodological difference between our study and Rankl et al. (2014) is our additional use of surface elevation change data and the annual resolution of the ITS_LIVE glacier surface velocity fields, which were intermittent (1992, 1993, 2003 and 2006–2013) in the study of Rankl et al. (2014). Many short-duration (median 2.3 years) Karakoram glacier surges (Fig. 11) may have occurred between the dates of velocity fields presented by Rankl et al. (2014), and the longer-lasting mass transfer signal of the same surges has only been picked up by our subsequent analyses of glacier surface elevation change. Also focusing on the Karakoram, Bhambri et al. (2017) compiled a list of 221 surge-type glaciers, 2 fewer than our estimate, but using a wide variety of observations spanning the period 1840–2015. As a broad por-

tion of the study period of Bhambri et al. (2017) covers the pre-satellite era, during which many short-duration or low-magnitude surges could have occurred undocumented, we would suggest that our inventory of surge-type glaciers provides a higher time-resolved estimate of the contemporary prevalence of surge-type glaciers in the Karakoram. Mukherjee et al. (2017) supplemented an inventory based on previously published results with surface elevation change datasets over the period 1964–2014 to identify 39 surge-type glaciers in the Tien Shan. Our results suggest there are 74 surge-type glaciers in the same study area. Again, the main methodological difference between our study and that of Mukherjee et al. (2017) is our use of the ITS_LIVE velocity time series, which enabled the identification of 35 additional surge-type glaciers. In contrast to the Pamirs, the substantial, long-term thinning experienced by glaciers in the Tien Shan (Bhattacharya et al., 2021) has likely dampened the signal of thickening caused by glacier surging in multi-decadal geodetic studies; thus, our additional examination of surface velocity anomalies has resulted in a more complete picture of the prevalence of surge-type glaciers here. The comparisons above emphasize the benefits of a multi-factor approach to identify surge-type glaciers, particularly in regions where glacier recession and mass loss rates are high. The availability of multi-temporal, globally resolved remotely sensed datasets should facilitate the assembly of extensive, accurate surge-type glaciers inventories in the future.

## 4.3 Distribution and geometry of surge and non-surge-type glaciers

We have studied the relationships between glacier geometry and surging behaviour. Our results are consistent with previous knowledge that surging glaciers systematically present greater areas, length and elevation range than non-surge-type glaciers (Clarke et al., 1986; Hamilton and Dowdeswell, 1996; Jiskoot et al., 1998, 2000, 2003; Barrand and Murray, 2006; Grant et al., 2009; Sevestre and Benn, 2015).

Greater lengths are correlated with increased surge-affected area in glacier complexes, with up to 45 % of total glacier area impacted in the Karakoram (Hispar Glacier). With surge-type glaciers being consistently in the higher end of the glacier area spectrum, such considerations need to be further studied as they are likely to have a significant impact on glacier mass balance (King et al., 2021) and glacier-related hazards (Kääb et al., 2021, for example) at the glacier scale.

While we observe lower median surface slope for surge-type glaciers, the results are not as clear-cut as for other studies such as Sevestre and Benn (2015) for example. The slight, non-statistically significant correlation between slope and surging described in our study is consistent with previous results in Yukon Territory (Clarke, 1991), Spitsbergen (Hamilton, 1992; Hamilton and Dowdeswell, 1996) and Karakoram (Bhambri et al., 2017). We however demonstrated that, in

HMA, longer glaciers with gentler surface slopes are more likely to be surge type than short steep glaciers. Length and slope thus act as independent controls on surging behaviour, a result that was first predicted in the enthalpy balance theory proposed by Benn et al. (2019).

We have shown that the aspect distributions for surge-type glaciers and non-surge-type glaciers do not substantially differ, which is consistent with previous knowledge on the studied regions (Bhambri et al., 2017; Mukherjee et al., 2017; Goerlich et al., 2020). Aspect distributions however differ in the Tibetan Mountains. We interpreted this result as the consequence of ice cap outlets outnumbering valley/mountain glaciers in this area, with the aspect of the latter being controlled by topography and geology, while the former presents less topographically constrained flow. From this, we argue that, in the Tibetan Mountains, aspect has no control over surging behaviour.

### 4.3.1 Impact of surges on mass balance

In Sect. 3.3, we presented glacier mass changes computed by the studies of Shean et al. (2020) and Hugonnet et al. (2021) with an emphasis on the potential differences in mass balance between surge-type and non-surge-type glaciers. In line with the results of Gardelle et al. (2013), Bolch et al. (2017) and Berthier and Brun (2019), we do not report any significant difference in mass budget between surge-type and non-surge-type glaciers over multi-decadal timescales, for all the greater HIMAP regions. We however described a marked increase in mass loss following surge cessation for two glaciers in the Karakoram. These results are consistent with the findings of Kochtitzky et al. (2019), King et al. (2021) and Bhattacharya et al. (2021). While such a mass loss is clearly identifiable in a minimal time frame after surge termination, it is likely averaged out over multi-decadal timescales. The regional distribution of mass changes over multi-decadal timescales is thus unlikely to be altered by surge-type behaviour, unless a significant number of glaciers surge repeatedly during the studied period (see Bhattacharya et al., 2021, for more) or in case of extreme events. Surges will however locally lead to increased variability in a glacier's meltwater runoff and hamper the water resource availability, potentially jeopardizing human livelihoods in the glacier's vicinity. The impact of surging over more localized scales thus needs to be further quantified. This requires the thorough study of a greater number of higher-temporal-resolution datasets to cover the entire surge cycle of a significant glacier sample and provide low-uncertainty mass balance estimates.

### 4.3.2 Temporal variability and velocities in surge active phase

In this study, we present distributions of surge active-phase duration derived from yearly ITS_LIVE velocity data. The moments of the presented distributions (here the percentiles)

**Table 4.** Comparison of the number of surge-type glaciers identified by other inventory studies in sub-regions of HMA with our results. d$T$: change in terminus position; d$V$: changes in glacier surface velocity; SF: surface features; d$H$: changes in glacier surface elevation.

| Study | Region | Period | Number of surge-type glaciers | This study | Data sources | Evidence |
|---|---|---|---|---|---|---|
| Bhambri et al. (2017) | Karakoram | 1840s–2017 | 221 | 223 | Landsat, ASTER, ground observations | d$T$, d$V$, SF |
| Copland et al. (2011) | – | 1960–2011 | 221 | 223 | Landsat, ASTER, Earth Resources Sat-1 | SF |
| Rankl et al. (2014) | – | 1976–2012 | 101 | – | Landsat, various synthetic aperture radar (SAR) images | d$T$, d$V$ |
| Sevestre and Benn (2015) | – | 1861–2013 | 106 | – | Literature | Various |
| Sevestre and Benn (2015) | Pamir | – | 820 | 223 | – | Various |
| Goerlich et al. (2020) | – | 1960s–2018 | 206 | – | Landsat, Corona KH-4, Hexagon KH-9, SRTM DEM, AW3D30 DEM | d$T$, d$H$ |
| Osipova et al. (1998), Kotlyakov et al. (2008) | – | 1973–2006 | 55 | – | Historical data, Resurs-F, ASTER | Various |
| Yasuda and Furuya (2015) | Western Kunlun | 1972–1992 | 9 | 60 | Landsat, various synthetic aperture radar (SAR) images | d$T$, d$V$ |
| Mukherjee et al. (2017) | Tien Shan | 1964–2014 | 39 | 74 | Landsat, Corona KH-4, Hexagon KH-9, Cartosat, SPOT | d$T$, d$H$ |
| Sevestre and Benn (2015) | – | 1861–2013 | 11 | – | Literature | Various |

are in line with previous studies focusing on each individual region (Quincey et al., 2011; Yasuda and Furuya, 2015; Quincey et al., 2015; Bhambri et al., 2017; Lv et al., 2019; Goerlich et al., 2020; Paul, 2020; Zhu et al., 2021). We nonetheless demonstrated that the differences between distinct HMA regions lie in the number of surges rather than the duration of individual surges. This result contrasts with the findings of Dowdeswell et al. (1991) stating that active phases in the Pamirs typically last for 1 to 2 years and disputes the claims of Zhu et al. (2021) that Tien Shan covers a greater range of active-phase duration than other regions in HMA.

We here further dispute the claim of Goerlich et al. (2020) that active-phase duration in the Pamirs is random. The presented results indeed demonstrate that the distribution of surge active-phase duration (both for HMA and the Pamirs) is more likely to be heavy-tailed or power-law-like.

### 4.4 On the physics of glacier surges

Our results suggest potential power-law-like and heavy-tailed distributions for a variety of surge-type glacier parameters. Using standard methods for power law validation, we most notably highlighted that the cumulative sum of IPR over the entire duration of surge active phases (as a crude proxy of energy dissipated during the active phase) potentially follows a power law distribution. The emergence of power-law-distributed quantities in avalanching geophysical systems (systems that suddenly release energy slowly accumulated over a period) arises from a wide variety of physical reasons (see Corral and González, 2019, for example), with the potential role of self-organized criticality

(Bak and Chen, 1991) in glacier surges being first discussed by Kavanaugh (2009). Given the importance of power-law-generating physical phenomena in understanding geophysical dynamical systems, we emphasize the need to validate the potential power laws described in this study with (1) complementary data (Sentinel 1 velocity data for example) providing velocity and surge active-phase duration estimates over a wider range of order of magnitudes and (2) adapting the existing glacier surge models proposed by Benn et al. (2019) or Thøgersen et al. (2019) to estimate the energy dissipated during the active phase of a surge.

### 5 Conclusions

In this paper, we presented a new inventory of surge-type glaciers for High Mountain Asia. This inventory is based on a multi-factor remote sensing approach, combining yearly velocity fields (ITS_LIVE), surface elevation change datasets published in prior studies and very-high-resolution satellite imagery (Bing Maps and Google Earth) to identify surge-type glaciers between 2000 and 2018. Overall, we identified 666 surge-type glaciers across HMA over the studied period, confirming 107 for which surging behaviour was believed "probable" or "possible". CE6 Compared to previously existing region-wide inventories, we newly identified 491 surge-type glaciers. We further studied the geometry of surge-type glaciers compared to non-surge-type glaciers and found relationships coherent with previous studies which have focused on smaller sub-samples of surge-type glaciers. We established a relationship between surging behaviour and glacier size, the most important being that for glacier complexes,

Please note the remarks at the end of the manuscript.

the area affected by surging is dependent on overall complex length following a power law relationship. The duration of the active phase of surge-type glaciers across HMA shows little variation in between greater HIMAP regions (median 2.6 ± 0.1 years). Finally, we defined the sum of velocity anomalies during a surge as a crude proxy for energy dissipated over the duration of the active phase and studied its distribution. We showed that this distribution is heavy-tailed and, using standard methods for power law validation, discussed that the sum of velocity anomalies and surge duration are potentially power law distributed. We recommend that our inventory serves as a baseline dataset in the further study of surge-type glacier characteristics.

*Data availability.* The inventory is available at https://doi.org/10.5281/zenodo.5524861 (Guillet et al., 2021) and through https://www.mountcryo.org/datasets/ (Mount Cryo, 2022).

*Author contributions.* GG, OK, and TB conceived the study. GG, OK, and ML generated the inventory, with support from SG. The code was developed by GG (velocity workflow and data analysis). GG and OK analysed the results with support from ML and DB. GG and OK wrote the paper and produced the figures, with contributions from all other co-authors.

*Competing interests.* At least one of the (co-)authors is a member of the editorial board of *The Cryosphere*. The peer-review process was guided by an independent editor, and the authors also have no other competing interests to declare.

*Acknowledgements.* We acknowledge the scientific editor, Etienne Berthier, as well as Jakob Steiner and one anonymous reviewer for the constructive remarks on the manuscript.

*Financial support.* This study was supported by the Strategic Priority Research Programs of the Chinese Academy of Sciences (grant nos. XDA20100300 and XDA19070202) and the Swiss National Science Foundation (200021E_177652/1) within the framework of the DFG Research Unit GlobalCDA (FOR2630).

*Review statement.* This paper was edited by Etienne Berthier and reviewed by Jakob Steiner and one anonymous referee.

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

## Remarks from the language copy-editor

CE1    Please confirm the slightly rephrased sentence.

CE2    We can only insert minor technical changes at this stage without approval from the editor. As the time periods used to generate the data examined would change, we need to request approval from the editor. Please provide an explanation for the requested changes.

CE3    Please confirm.

CE4    Please confirm the modified sentence.

CE5    It's our standard to use numerals for expressions used in a mathematical sense (ACS pg. 205).

CE6    I'm not sure what should be changed here as your comment wasn't clear. Please clarify.

## Remarks from the typesetter

TS1    Please provide date of last access.