# Peer review of "A regionally resolved inventory of High Mountain Asia surge-type glaciers, derived from a multi-factor remote sensing approach"

_The Cryosphere, 2021_

## Author Comment (AC1)

**Response to Reviewer 1**

THe authors would like to thank the reviewer for their careful reading and comments on our submitted manuscript.

In response to this feedback, we have made revisions to our manuscript following the reviewer's comments.

We now respond to the reviewer's comments point by point.

**General Comments**

1. There is one essential study that deals with controls of surges in HMA I am surprised you have left out of your Introduction but more importantly of your Discussion, namely (Barrand and Murray 2006). I am no co-author on that, so no bias, but they specifically tried to find controls, for the same variables that you investigated found similar results but also looked at other potential drivers. Added to that I am wondering, whether or why not you have considered to look at bed topography (simply via (Farinotti et al. 2019))? For some surging glaciers a particular shape of the valley/bed coincides with the transition from reservoir to receiving zone. That is not a request to do it, I am just wondering why you haven't expanded to other variables, like (Barrand and Murray 2006) also have done.

> We thank the reviewer for bringing this very important point forward.
> While we agree with the reviewer's comments, we wish to point out that the study from Barrand and Murray (2006) is mentionned in the Introduction. We further discussed our findings with the results of Barrand and Murray (2006) in Section 4.3.
> In comparison to the present work, Barrand and Murray propose to study two additional glacier attributes: glacier complexity and debris cover.
> Barrand and Murray (2006) numerically derive glacier complexity from the glacier permiter and area while other methods (Kienholz et al., 2013 for example) rely on semi-automatic approaches to identify the different glacier branches.
> Debris cover mapping similarly requires the use of advanced, problem-specific, algorithmic techniques.
> Here, we decided to keep our exploratory data analysis to attributes already existing in the RGI V6.0 and the velocity time series used.
> We nonetheless acknowledge that further investigations of surge-inducing parameters are necessary, and hope our inventory serves as a baseline for new studies.

2. The only really strong relation you find between variables is between glacier length and surge area (L215ff, and then Discussion). I am wondering however whether you did these comparisons also for the baseline 'non-surging' glaciers? Because I would assume that relation just holding true for any glacier, the longer it is the bigger it is and hence also the larger potential "surge area" it has. And

then that wouldn't really be a signal from the surging glaciers per se. Or am I missing something here?

> The "surge affected area" we refer to in the manuscript corresponds to the maximum surface area showing signs of surge activity (mostly increase in surface elevation) for each glacier, over the whole study period.
> As non-surge type glaciers do not display unstable behavior, they present no "surge-affected area".
> We here demonstrate that the relationship between a glacier's size, and the surface area destabilized during its most intense surge follows a power law.
> We thank the reviewer for bringing this very important point forward, and have made this clearer in the manuscript.

3. L230ff/L240ff: Since you do not address that further in the Discussion, I am curious why you think (a) there seems to be general smaller mass loss for surging glaciers but then later you show that they substantially loose mass at the end of the surge or just after it and (b) why you think that rapid onset of mass loss is? Just because ice mass was transported to lower elevations and hence melts faster? Or are there dynamic reasons at play that disguise some redistribution of mass as mass loss?

> The reasons for this sudden onset of mass loss are still partly understood.
> The study of Bhattacharya et al., (2021) however demonstrated similar increased mass loss for glaciers in the Ak-Shirak range following a period of synchronous surging, without any clear change in climate forcing.
> There is, to our knowledge, no evidence of dynamic thinning in the years following the termination of a surge. The down-glacier transfer of a significant volume of ice to lower elevation, and subsequent ice stagnation, appear to be reasonable hypotheses to explain the increased melt rate described in this manuscript.

4. L309ff: I think a major shortcoming of your study – that is naturally just stemming from the data we have access to – is that the period you investigate is shorter than some surge cycles. Khurdopin cycles have been around 20 years, Muchchuhar Glacier next to Shisper even much longer. So in a way you may miss some within that period. So when you compare to inventories that go further back I would have expected them to catch some surges which you may have failed to catch. But your numbers are consistently higher. That would suggest that your numbers are still too low (because you missed some of the currently quiescent ones). Considering that for example (Bhambri et al. 2017)'s data and maybe others as well are accessible, wouldn't it be prudent to compare your inventories and see which glaciers you agree on and where they find some you didn't and vice versa? (Bhambri et al. 2017) has a very similar number but you still say yours is 'more accurate'. From the evidence you provide I find that difficult to be sure of.

> We agree with the reviewer on the importance of this point and thank them for bringing it forward. The use of the terms "more accurate" was misleading and has been replaced in the manuscript.
> This section of the discussion specifically aims at comparing the propsoed HMA-wide

inventory to already existing regional ones.

As stated in the manuscript, our study differs from that of Bhambri et al., (2017), over several major points. First, Bhambri et al., (2017) do not consider surface elevaiton change datasets; this hampers one's ability to identify ice mass redistribution whether it is build up (reservoir zone) or increased mass loss (ablation zone) during the quiescent phase (see Fig 1.B South and Central Rimo examples in the manuscript) or anomalous mass gain in the ablation zone (see Fig 1.A).

Then, the identification of surges from Landsat scenes yields more space to interpretation than a primary scan of the quantified surface elevation changes (on which either surge or quiescence related signals are usually blatant) of all glaciers in the studied area ,and is thus more likely to lead to unobsvered surges.

Finally, our study relies on the use of the HIMAP regions proposed in Bolch et al. (2019) to define the different mountain ranges (called HIMAP regions in the manuscript). The HIMAP definition of the Karakoram differs to that used in Bhambri et al, (2017), and covers a greater area.

We further agree with the reviewer that the number of surge-type glaciers documented is likely underestimated due to the sampling rates of the surface velocity and elevation change datasets and the considered time period, as discussed in the manuscript.

5. As you are definitely aware, more recently a number of glaciers have been found to 'detach' rapidly (Kääb et al. 2021; Leinss et al. 2020). Have you made sure that these detachments are not classified as a surge by you? Even maybe for yet undetected detachments?

In the present work, we identified surge-type glaciers based on three distinct criteria. These criteria imply the use of datasets (surface elevation changes from DEM differences and mean yearly surface velocity) averaged over diverse time periods.

Signals resulting from sudden phenomena like the rapid detachments of low-angle glaciers are thus likely to be averaged-out by the dominant geophyscal signal.

As an example, no glacier studied in the works of Kaab et al., (2021) is in our present inventory.

Furthemore, the results form Kaab et al., (2021) explicitly describe sudden detachments on low-angle glaciers as the most dynamic end of the wide spectrum of surge-like instabilities thus further demonstrating that the various physical phenomena acting on surge-like glacier instabilities are still partly understood.

**Minor Comments**

1. L54: A study that actually looks at economic and infrastructure impacts of a surge, and that same surge, would be (Muhammad et al. 2021). I am a co-author on that study and I also do not think it is at all essential to cite here. I leave it up to you in case you find it helpful to make your point.

We have added the proposed reference as it is relevant for our manuscript, and thank the reviewer for bringing it to our knowledge.

2.L146: "Consortium et al. 2017" reads funny – I know citing the updated RGI is a bit strange but the official format is below, so I would at least go with 'RGI Consortium 2017'

> We have fixed this.

3. L155 and throughout: capitalize 'Glacier' when associated to a specific glacier. You do that sometimes, sometimes not. Same for 'Tibetan Plateau' on L181

> This has been fixed.

---

## Author Comment (AC2)

**Response to Reviewer 2**

**Response to Reviewer 2**

The authors would like to thank the Editor and reviewers for their careful reading and constructive comments on our first submission.

In response to this feedback, we have made major revisions to the manuscript, including significant additions. We hope that this version addresses the comments of the Editor and referees, and that our manuscript can now be accepted for publication.

We now respond, in more details, to the comments from Reviewer 2.

> L30: it would be useful to describe more explicitly as to which 'combinations of regional… and local… factors encourage instability', and in particular whether these are found in HMA

This has been done following reviewer's 1 recommendation.

> L43: change to 'distribution of surge-type glaciers…'

This has been fixed.

> L45: it's a bit debateable to state that 'no HMA-wide inventory of surge-type glacier exists', as one is already encompassed within the global study of Sevestre and Benn (2015). So I would refine the wording to say something like no 'dedicated' HMA-wide inventory currently exists.

We agree with the reviewer that this statement was debatable. We now have rephrased this sentence to mention the study of Sevestre and Benn (2015).

> L46: state which hazards are being referred to here – e.g., ice-dammed lakes? Perhaps also ice avalanches?

This has been fixed.

> L60: change to 'studies have documented…'

We have fixed this.

> L75: you say here that you identify surge-type glaciers from distinct widely used criteria, but the studies that you quote all use the presence of looped surface moraines as a major criteria, whereas it seems that you don't. It would be useful to add a sentence to make this clear, and to explain why you don't use this criteria.

The main aim of this study was to design an approach which could identify quantifiable changes in glaciological quantities associated with surging such as surface velocity and surface elevation. This methodology needed to be applicable to glaciers across HMA regardless of their surface characteristics (clean or debris-cover). Whilst we agree with the reviewer that incorporating the identification of geomorphological features, such as looped moraines or indeed a variety of other observable features (thrust-block/push moraine or ice strandlines) could be beneficial to validate the identification of surge-type behaviour, it would have limited our ability to conduct our analyses efficiently at a large scale as each glacier would have needed individual investigation to establish the presence of such features and their relation to surge behaviour. As a result, we chose not to incorporate the analyses of geomorphological features into our approach to identify surge-type glaciers.

> L78-80 (and elsewhere, e.g., L85): specify the time period that you're referring to for 'substantial and spatially concentrated surface elevation changes' and 'substantial variations in a glacier's velocity field' to be classified as indicative of surging

We have fixed this and now mention the typical time period considered.

> L97: I believe that the measurement periods for Hugonnet et al. (2021) are 2000-2004 and 2005-2009 (also check caption for Fig. 1, which should be 2010-2019, and elsewhere throughout your paper). See: http://maps.theia-land.fr/theia-cartographic-layers.html?year=2021&month=09&collection=glaciers

We thank the reviewer for bringing this point forward. We have modified the manuscript in agreement with the reviewer's suggestion.

> Fig. 2a: I assume that Fig. 2a shows the velocity patterns for a non-surging glacier, so make this clear in the caption. It would also help to highlight the positive heavy tail in Fig. 2b if you used a symmetrical x-axis scale for Fig. 2b (e.g., -150 to +150), to match the symmetrical scale already used for Fig.

We have fixed the figure's caption and now clearly mention the dynamical regime of each glacier. The symmetrical x-axis in Fig.2a results from the symmetry in the distribution. Representing the distribution of 2B on a similar symmetrical axis, would mean representing pixels that do not exist (from -20 to -150), while the shape of the distribution and the colorbar in Fig.2b highlight the disymmetry in that distribution.

> L139: provide the resolution in m that defines VHR

We have fixed this.

> L140: capitalize Bing Maps (and elsewhere, such as L161)

This has been fixed.

> L140: I don't know what 'infirm' means in this context; do you mean 'infer'?

We have now replaced infirm with rebut.

> L163: you provide the resolution of the velocity data here (240 m), but this should also be mentioned in Section 2.2

We now mention the resolution of ITS_LIVE yearly velocity datasets in Section 2.2

> L173: to avoid any potential ambiguity, I would suggest modifying this sentence to say something like 'at least 2 of 3 proposed identification criteria of rapid changes in surface elevation, surface velocity and surface crevassing' (assuming that these are the criteria that you're referring to here!)

We have rephrased this sentence using the reviewer's suggestion.

> L11 and L184: I'm not convinced by the statement that you newly identified 491 surge-type glaciers as it seems that you're only making comparisons with the RGI here? After working with the RGI myself, I know that their inventory is incomplete for their designation of which glaciers are surge-type. Rather, you need to make comparisons with other previously published studies to obtain an accurate number of which glaciers you've newly identified as surge-type, as you already do in Section 4.2 and Table 4. Indeed, Table 4 suggests that you haven't identified many new surge-type glaciers in some regions such as the Karakoram, and may have actually missed large numbers of them in the Pamirs.

We thank the reviewer for bringing this important forward.
We now have made an in-depth comparison between our inventory and the one proposed in Sevestre and Benn (2015) and found 490 newly identified surging glaciers.
We have added a substantial section within the Discussion which reads as follows :
"Sevestre and Benn (2015) proposed the only existing regional inventory of surge-type glaciers in HMA.
The number of surge-type glaciers documented in our inventory significantly differs from that of Sevestre and Benn (2015), especially in the Pamirs.
A further examination of the Sevestre and Benn (2015) inventory reveals that, out of the 827 surge-type glaciers documented, 284 correspond to individual tributaries within glacier complexes which are not individualized in the present study.
From the remaining 543 glaciers, 35 documented in the RGI V5.0 (on which Sevestre and Benn (2015) is based) do not exist in the RGI V6.0.
Furthermore, we found that the proposed inventory and the one from Sevestre and Benn (2015) only share 83 identified surge-type glaciers in the Pamirs.
This yields a difference of 390 surge-type glaciers between the two inventories.
Upon further examination of the remaining glacier population we note a median glacier area of $1.6km^2$, a sixth of the median area of the surge-type glacier population described in the present inventory ( $9.6km^2$).
Of those 390 in the Sevestre and Benn (2015) inventory, 30% present an area small than $1.0km^2$.
Close examination of these glaciers (Ujsu Glacier and the glaciers in its direct vicinity such as Aldzhaylau and Rakzou glaciers) using the surface elevation and surface velocity change data over the

period 2000-2018 did not yield any evidence of surge-type behavior.

We rather observed constant glacier mass loss and recession, with no clear signal of instability-related velocity anomalies.

Furthermore, no mention of such a high number of surge-type glaciers in the Pamirs can be found in the literature used by Sevestre and Benn (2015).

Kotlyakov et al (2008) indeed refer to Osipova et al (1998), mentioning "630 glaciers with indications of dynamic instability, 51 of them identified as surging", they later state that 55 surge-type glaciers had been documented up to 2006 in the Pamirs.

Given these unclear, conflicting and interpretive records of unstable glacier dynamics, the small sizes of glaciers in that population compared to other surge-type glaciers in the present and other studies (see next section), the lack of evidence for unstable behavior over the 2000-2018 period, as well as the new considerations laid by the enthalpy balance theory (Benn et al, 2019), we here question the surge-type behavior of this subset of 390 glaciers in the Pamirs.

Assuming we were to consider that population of glaciers as indeed surge-type, the estimated glacierized area covered by surge-type RGI polygons in HMA would become 20.6%."

> L199: to reinforce the point that there is high correlation between the geometrical parameters, it might be useful to state how longer glaciers by definition have shallower slopes if they cover the same elevation range as smaller glaciers.

We have rephrased the original sentence following the reviewer's suggestion.

> Fig. 5: please include part labels (a, b, c, d) for these figures, and indicate in the caption as to which geometrical attribute each figure part shows. It's currently a bit cryptic to try and figure out what log_range refers to, for example. Also provide units for each x-axis.

We have changed Figure 5 according to the reviewer's suggestion.

> L235: I dislike sentences with clauses in brackets as it makes them difficult to follow, particularly when there are multiple such sentences back-to-back. It takes about the same amount of space to write out the sentences properly, but makes them easier to understand: e.g., 'We however note that balance distribution for surge-type glaciers in the Karakoram is positively skewed with g = 1.1, and negatively skewed with g = −1.4 for non surge-type glaciers.'

We have rephrased the paragraph following the reviewer's comment.

> L238: I don't follow the comment that the Himalayas are not representative with <10 surge-type glaciers, as Fig. 7 specifies that the Himalayas has n=13 surge-type glaciers, and Table 1 suggests n=14 (unless Himalayas is defined in a different way in Fig. 10 than in other figs and tables, which relates to my comments below)

This typo has been fixed.

> Fig. 7 and 8: please clarify how the six regions here compare to the HIMAP regions listed in Table 1 (perhaps by adding an extra column to Table 1?). For example, I can't figure out which of the six regions the Gangdise Mountains fits into. Some numbers also seem to be inconsistent between Table 1 and Figure 7: e.g., Central + Western + Eastern Himalaya = 14 in Table 1, but n = 13 in Fig. 7. Also present the figure parts in the same order in each figure so that it's easier to compare them (e.g., Tien Shan is shown first in Fig. 7, but Tibet is shown first in Fig. 8)

We have fixed this following the reviewer's comments.

> Fig. 9 and 10: these use 8 regions, compared to the 6 regions in Figs. 7 and 8, and 22 regions in Table 1. This makes it essentially impossible to make comparisons between the different figures, and makes it even more confusing as to which regions in Table 1 are included in which regions in the figures. Please be consistent throughout, and clearly define how the regions relate to each other.

We agree with the reviewer that regular switches between the use of greater HIMAP and standard HIMAP regions was misleading and affected the manuscript's clarity.
Following the reviewer's suggestion, we have added a new column to Table 1 to clarify how regions relate to each other.

> L242 (and elsewhere): formal glacier names should be capitalized when referring to a single glacier: e.g., Khurdopin Glacier, Hispar Glacier

We have fixed this.

> L244: add superscript -1 at the end of: ' -0.22 – 0.3 m w.e. a-1'

This has been fixed.

> Fig. 11: I don't follow the x-axis label for parts a and b: how can the units be both dimensionless and in m yr-1?

This was meant to remind the reader with the original unit of the datasets used in this figure. We however have removed the unit.

> L260: I've read little, if anything, about previous surges lasting for 18 years in the HMA, so it would be useful to expand on this to provide more information about the location and characteristics of these, and how you can be sure that they surged for that entire time. Presumably it would also be more accurate to say 'at least 18 years', since this is the maximum length of your record?

The surge-type glaciers documented in this inventory are represented using RGI polygons, as tributaries are not individualized. The glaciers displaying surges as long as 18 years hence are glacier complexes showing sequential surges of their different tributaries.

We further wish to mention that the works from King et al. (2021) document surges in the Geladandong involving elevated velocities for ~15 years.

> L261: Is the equation reference here correct? IPR is defined in Equation 3, while Equation 4 defines the surge index.

The reference to Equation 3 has been corrected.

> L271: 'active phase' is repeated twice here

This has been fixed

> L278: change to 'prevents identification of active phases…'

We have fixed this.

> L286-7: this statement is a bit meaningless without anything to back it up; instead, it would be useful to provide some specific numbers here to convince the reader that the patterns you measured are real. For example, provide an average value for the quoted elevation uncertainty in the Hugonnet dataset, and state what the average elevation changes were that you measured on surge-type glaciers. Same for velocity changes.

We thank the reviwer for bringing this point forward. We have provided an average value both for the quoted elevation uncertainty and the quoted surface velocity uncertainty.
The secion has been entirely reworked and know reads as follows :
"'The surge-type glacier identification criteria which we have followed are built on a number of different publicly available datasets generated in previous studies.
Each of these datasets are imperfect representation of real geophysical signals and attributes and present a level of uncertainty.
The error associated with the elevation change datasets of Hugonnet et al (2021)} varies somewhat between different time periods.
Between 2000-2005 and 2005-2010 the mean error over glacier surfaces in our inventory is 1.04 m a-1 and 0.97 m a-1, respectively. The mean error over glacier surfaces in our inventory using data from Brun et al (2017) is 0.43 m a-1, although we note lower coverage of this elevation change dataset over higher reaches of glaciers, where associated error would also be higher.
Error estimates associated with the elevation change dataset of Shean et al (2020) are not freely available, but we would expect similar levels of error to be associated with these data to those of Hugonnet et al (2021). Shean et al (2020) estimate similar levels of uncertainty alongside their geodetic glacier mass balance estimates, and as surface elevation change error tends to dominate the geodetic mass balance budget, the error associated with the underlying elevation change data should therefore be similar.
Importantly, the error associated with elevation change datasets over glacier tongues is predominantly well below the magnitude of the changes associated with surge behaviour (up to 10 m a-1).
The error associated with the surface velocity data similarly varies between different periods, due to the

switch from Landsat7 to Landsat 8 imagery.

Between 2000-2013 and 2013-2018 the median error in glacier surface velocity in the inventory is around 1.35 m a-1 and 0.37 m a-1 respectively.

The maximum error is significantly larger for 2000-2013 than 2013-2018, with 63 m a-1 and 11.3 m a-1. Errors in the ITS_LIVE surface velocity dataset typically cluster in the accumulation area of glaciers, where trackable features are less abundant.

The use of error-weighted surface velocity fields (Section 2.2) ensures a greater penalization of aberrant surface velocity pixels in the accumulation area.

In the ablation area and over glacier tongues, errors on surface velocity lie well below the magnitude of changes in velocity resulting from a surge (up to 150 m a-1)."'

> L296: remind the reader here of what the discriminatory criteria are, so that they don't have to go searching back through the previous sections

As suggested by the reviewer, we now remind the reader of the diagnostic criteria.

> L302: change to 'certain level of certainty'

This has been fixed.

> L315: you also need to make comparisons with the inventory of Sevestre and Benn (2015) here. You currently do this in section 4.3, but that text would be better moved to here. It would also be helpful to discuss why Sevestre and Benn (2015) identify so many more surge-type glaciers in the Pamirs than you, even after removing their duplicates.

We agree with the reviewer on the importance of a comparison between our inventory and that of Sevestre and Benn (2015). As mentioned earlier, we have added such a discussion where the reviewer suggested it.

> L322: you're missing the inventory of Copland et al. (2011), who identified 90 surge-type glaciers in the Karakoram

We thank the reviewer for bringing this important reference to our atention. It has been added to our manuscript following the reviewer's suggestion.

> L350: change to 'up to 45% of total glacier area…' to make it clear that you're referring to the entire glaciated region here, and not just the proportion impacted on each individual surge-type glacier

We have fixed this using the reviewe'rs suggestion.

> L372: seem to be missing some words here? 'in mass balance a single…' doesn't make sense as written

We have rephrased this sentence.

---

## Author Response (AR2)

**Response to Editor and Reviewers**

The authors would like to thank the Editor and reviewers for their careful reading and constructive comments on our first submission.

In response to this feedback, we have made additional minor revisions to the manuscript, including figure reworks. We hope that this version addresses the comments of the Editor and referees, and that our manuscript can now be accepted for publication.

We now respond, in more details, to the comments from both reviewers.

**Response to Editor**

> In your answer to reviewer#1, you did not clarify why you did not use bed topography. The reviewer stated "Added to that I am wondering, whether or why not you have considered to look at bed topography (simply via (Farinotti et al. 2019))? For some surging glaciers a particular shape of the valley/bed coincides with the transition from reservoir to receiving zone.". Your choice makes sense to me given uncertainties in this dataset but this reviewer would certainly be satisfied to receive an answer to this specific aspect.

We thank the Editor for pointing this omission out to us. We apoligize for this and further adress this point raised by Reviewer 1 in the present Response.

> L10. A reader may be surprised that 107 and 491 do not sum up to 666. Reason for this difference? (I found the answer later in the text but making the abstract crystal clear would be best)

As suggested by the Editor here, and further in his comments to our manuscript, we now have added a sentence mentionning that 68 glaciers were already classified as surge-type in the RGI V6.0.

> L87. Maybe this would be the place to indicate that loop moraines or others similar surface features are not considered as criteria in this study, in response to a comment by reviewer#2.

Following the Editor's suggestion, we have added such a statement.

> L105. Space between the number and its unit.

This has been fixed.

> L105. I suggest using "were" instead of "are" to underline that this was not done in this study but earlier, by others.

We agree with the Editor that this sentence lacked clarity. It has been modified using the Editor's suggestion.

> Figure 2. Can you tell in the legend for which year i, the map/distribution refer to?

This has been fixed.

> Figure 2. Authors could add two additional panels with the absolute values of Dv and indicate in the plot P50 and P95 to make it clear to all how IPR (and then the surge index) is obtained. Value of IPR and the surge index could be written on the new panels for these two glaciers. This is a suggestion, not mandatory.

We thank the Editori for his suggestion.
We have however decided not to change the figure at this stage of the revision process.

> L167. Why 300 out of 666 was unclear to me. Is it because of the 5 km² threshold? If so, make it clearer. Otherwise better explain.

We have made this clear that 300 out of 666 refers to glacier complexes, as glacier tributaries are not individualized in the RGI V6.0.

> L174. Space between Section and 2.1

This has been fixed.

> L176. This is my most substantial comment: authors have produced these two sub-inventories. But never compared them. A few words about the number of glaciers in each of them and the number of common ones would be welcome. Due to the noise in the velocity data one can expect quite some "noise" in the identification of the glaciers based on the IPR so it would be useful to provide a bit more detail here (or later in the result section).

Following the Editor's suggestions, we now have added a paragraph in our "Results" section comparing the two sub-inventories in greater details.

> L190. Here I understand the numbers in the abstract (my comment L10 above). Maybe add 68 "already identified" in the abstract?

As stated earlier, we have modified the abstract following the Editor's comments.

> Figure 5. For better use of space in the journal, I think a lot of blank areas could be removed from the figure. Especially in the vertical direction. Each panel does not need to be square.

We agree with the Editor that the amount of blanks in this figure lead to space loss.
We have reworked this figure following the Editor's suggestion.

> L239. Clarify that this is the glacier-wide mass balance.

We have now clarified this.

> L246. "the mass of the distribution" is maybe not the best term for a section dealing with "mass balance".

We agree with the Editor that in this particular case, the similarity between mathematical and glaciological terms hampers the readability of the manuscript.
We now refer to "the density of the distribution".

> L253. "-1" as exponent

This has been fixed.

> Table 2. can you clarify the source of the mass balance data in this Table? If this is from Hugonnet et al., 2021 I would be careful to indicate that their results have large uncertainties for annual time steps. They did not resolve well the annual variability and extreme mass balance years are smoothed out by the gaussian processes. To keep in mind.

We agree with the reviewer that uncertainties in the mass balance datasets had to be better mentionned. Following the Editor's suggestions, we now have added the uncertainties associated to each surface elevation change estimate in Table 2 and clearly refer to the Hugonnet et al., 2021 dataset in the caption of the table.

> Figure 10. Maybe the text should stress some differences between both studies (Shean vs. Hugonnet) even for the same sample of glaciers and almost similar time period. It makes detection of a specific mass balance behaviour of surge-type glaciers tricky given uncertainties in the mass balance dataset.

This has been fixed.

> L261. provide the unit (years)

We have fixed this.

> L345. Indicate that this is in the Sevestre and Benn, 2015 inventory.

We now clearly refer to Sevestre and Benn (2015).

> L426. I would insist here on the need for high temporal resolution data. Right now this is not the case in Hugonnet et al. dataset because ASTER acquisitions were scarce and their uncertainties rather high.

Following the reviewers'

> L433. "that rather than "than"

This has been fixed.

**Response to Reviewer 1**

**General Comments**

> There is one essential study that deals with controls of surges in HMA I am surprised you
> have left out of your Introduction but more importantly of your Discussion, namely (Barrand
> and Murray 2006). I am no co-author on that, so no bias, but they specifically tried to find
> controls, for the same variables that you investigated found similar results but also looked at
> other potential drivers. Added to that I am wondering, whether or why not you have
> considered to look at bed topography (simply via (Farinotti et al. 2019))? For some surging
> glaciers a particular shape of the valley/bed coincides with the transition from reservoir to
> receiving zone. That is not a request to do it, I am just wondering why you haven't expanded
> to other variables, like (Barrand and Murray 2006) also have done.

We thank the reviewer for bringing this very important point forward.
While we agree with the reviewer's comments, we wish to point out that the study from Barrand and
Murray (2006) is mentionned in the Introduction. We further discussed our findings with the results of
Barrand and Murray (2006) in Section 4.3.
In comparison to the present work, Barrand and Murray propose to study two additional glacier
attributes: glacier complexity and debris cover.
Barrand and Murray (2006) numerically derive glacier complexity from the glacier permiter and area
while other methods (Kienholz et al., 2013 for example) rely on semi-automatic approaches to identify
the different glacier branches.
Debris cover mapping similarly requires the use of advanced, problem-specific, algorithmic techniques.
It is unclear what Farinotti et al., (2019) paper the reviewer refers to.
Assuming the reviewer suggests to use the Farinotti et al., (2019) ice thickness dataset to invert the
bed topography, we argue that using a global scale dataset -presenting high regional uncertainties- as
input within an ill-posed inverse problem to compute a single lies beyond the scope of the present
manuscript.
Here, we decided to keep our exploratory data analysis to attributes already existing in the RGI V6.0
and the velocity time series used.
We nonetheless acknowledge that further investigations of surge-inducing parameters are necessary,
and hope our inventory serves as a baseline for new studies.

> The only really strong relation you find between variables is between glacier length and surge
> area (L215ff, and then Discussion). I am wondering however whether you did these
> comparisons also for the baseline 'non-surging' glaciers? Because I would assume that
> relation just holding true for any glacier, the longer it is the bigger it is and hence also the
> larger potential "surge area" it has. And then that wouldn't really be a signal from the surging
> glaciers per se. Or am I missing something here?

The "surge affected area" we refer to in the manuscript corresponds to the maximum surface area showing signs of surge activity (mostly increase in surface elevation) for each glacier, over the whole study period.

As non-surge type glaciers do not display unstable behavior, they present no "surge-affected area".

We here demonstrate that the relationship between a glacier's size, and the surface area destabilized during its most intense surge follows a power law.

We thank the reviewer for bringing this very important point forward, and have made this clearer in the manuscript.

> L230ff/L240ff: Since you do not address that further in the Discussion, I am curious why you think (a) there seems to be general smaller mass loss for surging glaciers but then later you show that they substantially loose mass at the end of the surge or just after it and (b) why you think that rapid onset of mass loss is? Just because ice mass was transported to lower elevations and hence melts faster? Or are there dynamic reasons at play that disguise some redistribution of mass as mass loss?

The reasons for this sudden onset of mass loss are still partly understood.

The study of Bhattacharya et al., (2021) however demonstrated similar increased mass loss for glaciers in the Ak-Shirak range following a period of synchronous surging, without any clear change in climate forcing.

There is, to our knowledge, no evidence of dynamic thinning in the years following the termination of a surge. The down-glacier transfer of a significant volume of ice to lower elevation, and subsequent ice stagnation, appear to be reasonable hypotheses to explain the increased melt rate described in this manuscript.

> L309ff: I think a major shortcoming of your study – that is naturally just stemming from the data we have access to – is that the period you investigate is shorter than some surge cycles. Khurdopin cycles have been around 20 years, Muchchuhar Glacier next to Shisper even much longer. So in a way you may miss some within that period. So when you compare to inventories that go further back I would have expected them to catch some surges which you may have failed to catch. But your numbers are consistently higher. That would suggest that your numbers are still too low (because you missed some of the currently quiescent ones). Considering that for example (Bhambri et al. 2017)'s data and maybe others as well are accessible, wouldn't it be prudent to compare your inventories and see which glaciers you agree on and where they find some you didn't and vice versa? (Bhambri et al. 2017) has a very similar number but you still say yours is 'more accurate'. From the evidence you provide I find that difficult to be sure of.

We agree with the reviewer on the importance of this point and thank them for bringing it forward. The use of the terms "more accurate" was misleading and has been replaced in the manuscript.

This section of the discussion specifically aims at comparing the propsoed HMA-wide inventory to already existing regional ones.

As stated in the manuscript, our study differs from that of Bhambri et al., (2017), over several major points. First, Bhambri et al., (2017) do not consider surface elevaiton change datasets; this hampers

one's ability to identify ice mass redistribution whether it is build up (reservoir zone) or increased mass loss (ablation zone) during the quiescent phase (see Fig 1.B South and Central Rimo examples in the manuscript) or anomalous mass gain in the ablation zone (see Fig 1.A).

Then, the identification of surges from Landsat scenes yields more space to interpretation than a primary scan of the quantified surface elevation changes (on which either surge or quiescence related signals are usually blatant) of all glaciers in the studied area ,and is thus more likely to lead to unobsvered surges.

Finally, our study relies on the use of the HIMAP regions proposed in Bolch et al. (2019) to define the different mountain ranges (called HIMAP regions in the manuscript). The HIMAP definition of the Karakoram differs to that used in Bhambri et al, (2017), and covers a greater area.

We further agree with the reviewer that the number of surge-type glaciers documented is likely underestimated due to the sampling rates of the surface velocity and elevation change datasets and the considered time period, as discussed in the manuscript.

> As you are definitely aware, more recently a number of glaciers have been found to 'detach' rapidly (Kääb et al. 2021; Leinss et al. 2020). Have you made sure that these detachments are not classified as a surge by you? Even maybe for yet undetected detachments?

In the present work, we identified surge-type glaciers based on three distinct criteria.

These criteria imply the use of datasets (surface elevation changes from DEM differences and mean yearly surface velocity) averaged over diverse time periods.

Signals resulting from sudden phenomena like the rapid detachments of low-angle glaciers are thus likely to be averaged-out by the dominant geophyscal signal.

As an example, no glacier studied in the works of Kaab et al., (2021) is in our present inventory. Furthemore, the results form Kaab et al., (2021) explicitly describe sudden detachments on low-angle glaciers as the most dynamic end of the wide spectrum of surge-like instabilities thus further demonstrating that the various physical phenomena acting on surge-like glacier instabilities are still partly understood.

**Minor Comments**

> L54: A study that actually looks at economic and infrastructure impacts of a surge, and that same surge, would be (Muhammad et al. 2021). I am a co-author on that study and I also do not think it is at all essential to cite here. I leave it up to you in case you find it helpful to make your point.

We have added the proposed reference as it is relevant for our manuscript, and thank the reviewer for bringing it to our knowledge.

> L146: "Consortium et al. 2017" reads funny – I know citing the updated RGI is a bit strange but the official format is below, so I would at least go with 'RGI Consortium 2017'

We have fixed this.

> L155 and throughout: capitalize 'Glacier' when associated to a specific glacier. You do that sometimes, sometimes not. Same for 'Tibetan Plateau' on L181

This has been fixed.

**Response to Reviewer 2**

> L30: it would be useful to describe more explicitly as to which 'combinations of regional… and local… factors encourage instability', and in particular whether these are found in HMA

This has been done following reviewer's 1 recommendation.

> L43: change to 'distribution of surge-type glaciers…'

This has been fixed.

> L45: it's a bit debateable to state that 'no HMA-wide inventory of surge-type glacier exists', as one is already encompassed within the global study of Sevestre and Benn (2015). So I would refine the wording to say something like no 'dedicated' HMA-wide inventory currently exists.

We agree with the reviewer that this statement was debatable. We now have rephrased this sentence to mention the study of Sevestre and Benn (2015).

> L46: state which hazards are being referred to here – e.g., ice-dammed lakes? Perhaps also ice avalanches?

This has been fixed.

> L60: change to 'studies have documented…'

We have fixed this.

> L75: you say here that you identify surge-type glaciers from distinct widely used criteria, but the studies that you quote all use the presence of looped surface moraines as a major criteria, whereas it seems that you don't. It would be useful to add a sentence to make this clear, and to explain why you don't use this criteria.

The main aim of this study was to design an approach which could identify quantifiable changes in glaciological quantities associated with surging such as surface velocity and surface elevation. This methodology needed to be applicable to glaciers across HMA regardless of their surface characteristics (clean or debris-cover). Whilst we agree with the reviewer that incorporating the identification of geomorphological features, such as looped moraines or indeed a variety of other observable features (thrust-block/push moraine or ice strandlines) could be beneficial to validate the identification of surge-type behaviour, it would have limited our ability to conduct our analyses efficiently at a large scale as each glacier would have needed individual investigation to establish the presence of such features and

their relation to surge behaviour. As a result, we chose not to incorporate the analyses of geomorphological features into our approach to identify surge-type glaciers.

> L78-80 (and elsewhere, e.g., L85): specify the time period that you're referring to for 'substantial and spatially concentrated surface elevation changes' and 'substantial variations in a glacier's velocity field' to be classified as indicative of surging

We have fixed this and now mention the typical time period considered.

> L97: I believe that the measurement periods for Hugonnet et al. (2021) are 2000-2004 and 2005-2009 (also check caption for Fig. 1, which should be 2010-2019, and elsewhere throughout your paper). See: http://maps.theia-land.fr/theia-cartographic-layers.html?year=2021&month=09&collection=glaciers

We thank the reviewer for bringing this point forward. We have modified the manuscript in agreement with the reviewer's suggestion.

> Fig. 2a: I assume that Fig. 2a shows the velocity patterns for a non-surging glacier, so make this clear in the caption. It would also help to highlight the positive heavy tail in Fig. 2b if you used a symmetrical x-axis scale for Fig. 2b (e.g., -150 to +150), to match the symmetrical scale already used for Fig.

We have fixed the figure's caption and now clearly mention the dynamical regime of each glacier. The symmetrical x-axis in Fig.2a results from the symmetry in the distribution. Representing the distribution of 2B on a similar symmetrical axis, would mean representing pixels that do not exist (from -20 to -150), while the shape of the distribution and the colorbar in Fig.2b highlight the disymmetry in that distribution.

> L139: provide the resolution in m that defines VHR

We have fixed this.

> L140: capitalize Bing Maps (and elsewhere, such as L161)

This has been fixed.

> L140: I don't know what 'infirm' means in this context; do you mean 'infer'?

We have now replaced infirm with rebut.

> L163: you provide the resolution of the velocity data here (240 m), but this should also be mentioned in Section 2.2

We now mention the resolution of ITS_LIVE yearly velocity datasets in Section 2.2

> L173: to avoid any potential ambiguity, I would suggest modifying this sentence to say something like 'at least 2 of 3 proposed identification criteria of rapid changes in surface elevation, surface velocity and surface crevassing' (assuming that these are the criteria that you're referring to here!)

We have rephrased this sentence using the reviewer's suggestion.

> L11 and L184: I'm not convinced by the statement that you newly identified 491 surge-type glaciers as it seems that you're only making comparisons with the RGI here? After working with the RGI myself, I know that their inventory is incomplete for their designation of which glaciers are surge-type. Rather, you need to make comparisons with other previously published studies to obtain an accurate number of which glaciers you've newly identified as surge-type, as you already do in Section 4.2 and Table 4. Indeed, Table 4 suggests that you haven't identified many new surge-type glaciers in some regions such as the Karakoram, and may have actually missed large numbers of them in the Pamirs.

We thank the reviewer for bringing this important forward.

We now have made an in-depth comparison between our inventory and the one proposed in Sevestre and Benn (2015) and found 490 newly identified surging glaciers.

We have added a substantial section within the Discussion which reads as follows :

"Sevestre and Benn (2015) proposed the only existing regional inventory of surge-type glaciers in HMA.

The number of surge-type glaciers documented in our inventory significantly differs from that of Sevestre and Benn (2015), especially in the Pamirs.

A further examination of the Sevestre and Benn (2015) inventory reveals that, out of the 827 surge-type glaciers documented, 284 correspond to individual tributaries within glacier complexes which are not individualized in the present study.

From the remaining 543 glaciers, 35 documented in the RGI V5.0 (on which Sevestre and Benn (2015) is based) do not exist in the RGI V6.0.

Furthermore, we found that the proposed inventory and the one from Sevestre and Benn (2015) only share 83 identified surge-type glaciers in the Pamirs.

This yields a difference of 390 surge-type glaciers between the two inventories.

Upon further examination of the remaining glacier population we note a median glacier area of $1.6 km^2$, a sixth of the median area of the surge-type glacier population described in the present inventory ($9.6 km^2$).

Of those 390 in the Sevestre and Benn (2015) inventory, 30% present an area small than $1.0 km^2$.

Close examination of these glaciers (Ujsu Glacier and the glaciers in its direct vicinity such as Aldzhaylau and Rakzou glaciers) using the surface elevation and surface velocity change data over the period 2000-2018 did not yield any evidence of surge-type behavior.

We rather observed constant glacier mass loss and recession, with no clear signal of instability-related velocity anomalies.

Furthermore, no mention of such a high number of surge-type glaciers in the Pamirs can be found in the literature used by Sevestre and Benn (2015).

Kotlyakov et al (2008) indeed refer to Osipova et al (1998), mentioning "630 glaciers with indications of dynamic instability, 51 of them identified as surging", they later state that 55 surge-type glaciers had

been documented up to 2006 in the Pamirs.

Given these unclear, conflicting and interpretive records of unstable glacier dynamics, the small sizes of glaciers in that population compared to other surge-type glaciers in the present and other studies (see next section), the lack of evidence for unstable behavior over the 2000-2018 period, as well as the new considerations laid by the enthalpy balance theory (Benn et al, 2019), we here question the surge-type behavior of this subset of 390 glaciers in the Pamirs.

Assuming we were to consider that population of glaciers as indeed surge-type, the estimated glacierized area covered by surge-type RGI polygons in HMA would become 20.6%."

> L199: to reinforce the point that there is high correlation between the geometrical parameters, it might be useful to state how longer glaciers by definition have shallower slopes if they cover the same elevation range as smaller glaciers.

We have rephrased the original sentence following the reviewer's suggestion.

> Fig. 5: please include part labels (a, b, c, d) for these figures, and indicate in the caption as to which geometrical attribute each figure part shows. It's currently a bit cryptic to try and figure out what log_range refers to, for example. Also provide units for each x-axis.

We have changed Figure 5 according to the reviewer's suggestion.

> L235: I dislike sentences with clauses in brackets as it makes them difficult to follow, particularly when there are multiple such sentences back-to-back. It takes about the same amount of space to write out the sentences properly, but makes them easier to understand: e.g., 'We however note that balance distribution for surge-type glaciers in the Karakoram is positively skewed with g = 1.1, and negatively skewed with g = −1.4 for non surge-type glaciers.'

We have rephrased the paragraph following the reviewer's comment.

> L238: I don't follow the comment that the Himalayas are not representative with <10 surge-type glaciers, as Fig. 7 specifies that the Himalayas has n=13 surge-type glaciers, and Table 1 suggests n=14 (unless Himalayas is defined in a different way in Fig. 10 than in other figs and tables, which relates to my comments below)

This typo has been fixed.

> Fig. 7 and 8: please clarify how the six regions here compare to the HIMAP regions listed in Table 1 (perhaps by adding an extra column to Table 1?). For example, I can't figure out which of the six regions the Gangdise Mountains fits into. Some numbers also seem to be inconsistent between Table 1 and Figure 7: e.g., Central + Western + Eastern Himalaya = 14 in Table 1, but n = 13 in Fig. 7. Also present the figure parts in the same order in each figure so that it's easier to compare them (e.g., Tien Shan is shown first in Fig. 7, but Tibet is shown first in Fig. 8)

We have fixed this following the reviewer's comments.

> Fig. 9 and 10: these use 8 regions, compared to the 6 regions in Figs. 7 and 8, and 22 regions in Table 1. This makes it essentially impossible to make comparisons between the different figures, and makes it even more confusing as to which regions in Table 1 are included in which regions in the figures. Please be consistent throughout, and clearly define how the regions relate to each other.

We agree with the reviewer that regular switches between the use of greater HIMAP and standard HIMAP regions was misleading and affected the manuscript's clarity.
Following the reviewer's suggestion, we have added a new column to Table 1 to clarify how regions relate to each other.

> L242 (and elsewhere): formal glacier names should be capitalized when referring to a single glacier: e.g., Khurdopin Glacier, Hispar Glacier

We have fixed this.

> L244: add superscript -1 at the end of: ' -0.22 – 0.3 m w.e. a-1'

This has been fixed.

> Fig. 11: I don't follow the x-axis label for parts a and b: how can the units be both dimensionless and in m yr-1?

This was meant to remind the reader with the original unit of the datasets used in this figure.
We however have removed the unit.

> L260: I've read little, if anything, about previous surges lasting for 18 years in the HMA, so it would be useful to expand on this to provide more information about the location and characteristics of these, and how you can be sure that they surged for that entire time. Presumably it would also be more accurate to say 'at least 18 years', since this is the maximum length of your record?

The surge-type glaciers documented in this inventory are represented using RGI polygons, as tributaries are not individualized. The glaciers displaying surges as long as 18 years hence are glacier complexes showing sequential surges of their different tributaries.
We further wish to mention that the works from King et al. (2021) document surges in the Geladandong involving elevated velocities for ~15 years.

> L261: Is the equation reference here correct? IPR is defined in Equation 3, while Equation 4 defines the surge index.

The reference to Equation 3 has been corrected.

> L271: 'active phase' is repeated twice here

This has been fixed

> L278: change to 'prevents identification of active phases…'

We have fixed this.

> L286-7: this statement is a bit meaningless without anything to back it up; instead, it would be useful to provide some specific numbers here to convince the reader that the patterns you measured are real. For example, provide an average value for the quoted elevation uncertainty in the Hugonnet dataset, and state what the average elevation changes were that you measured on surge-type glaciers. Same for velocity changes.

We thank the reviwer for bringing this point forward. We have provided an average value both for the quoted elevation uncertainty and the quoted surface velocity uncertainty.
The secion has been entirely reworked and know reads as follows :
'''The surge-type glacier identification criteria which we have followed are built on a number of different publicly available datasets generated in previous studies.
Each of these datasets are imperfect representation of real geophysical signals and attributes and present a level of uncertainty.
The error associated with the elevation change datasets of Hugonnet et al (2021)} varies somewhat between different time periods.
Between 2000-2005 and 2005-2010 the mean error over glacier surfaces in our inventory is 1.04 m a-1 and 0.97 m a-1, respectively. The mean error over glacier surfaces in our inventory using data from Brun et al (2017) is 0.43 m a-1, although we note lower coverage of this elevation change dataset over higher reaches of glaciers, where associated error would also be higher.
Error estimates associated with the elevation change dataset of Shean et al (2020) are not freely available, but we would expect similar levels of error to be associated with these data to those of Hugonnet et al (2021). Shean et al (2020) estimate similar levels of uncertainty alongside their geodetic glacier mass balance estimates, and as surface elevation change error tends to dominate the geodetic mass balance budget, the error associated with the underlying elevation change data should therefore be similar.
Importantly, the error associated with elevation change datasets over glacier tongues is predominantly well below the magnitude of the changes associated with surge behaviour (up to 10 m a-1).
The error associated with the surface velocity data similarly varies between different periods, due to the switch from Landsat7 to Landsat 8 imagery.
Between 2000-2013 and 2013-2018 the median error in glacier surface velocity in the inventory is around 1.35 m a-1 and 0.37 m a-1 respectively.
The maximum error is significantly larger for 2000-2013 than 2013-2018, with 63 m a-1 and 11.3 m a-1.
Errors in the ITS_LIVE surface velocity dataset typically cluster in the accumulation area of glaciers, where trackable features are less abundant.
The use of error-weighted surface velocity fields (Section 2.2) ensures a greater penalization of aberrant surface velocity pixels in the accumulation area.

In the ablation area and over glacier tongues, errors on surface velocity lie well below the magnitude of changes in velocity resulting from a surge (up to 150 m a-1).'"

> L296: remind the reader here of what the discriminatory criteria are, so that they don't have to go searching back through the previous sections

As suggested by the reviewer, we now remind the reader of the diagnostic criteria.

> L302: change to 'certain level of certainty'

This has been fixed.

> L315: you also need to make comparisons with the inventory of Sevestre and Benn (2015) here. You currently do this in section 4.3, but that text would be better moved to here. It would also be helpful to discuss why Sevestre and Benn (2015) identify so many more surge-type glaciers in the Pamirs than you, even after removing their duplicates.

We agree with the reviewer on the importance of a comparison between our inventory and that of Sevestre and Benn (2015). As mentioned earlier, we have added such a discussion where the reviewer suggested it.

> L322: you're missing the inventory of Copland et al. (2011), who identified 90 surge-type glaciers in the Karakoram

We thank the reviewer for bringing this important reference to our atention. It has been added to our manuscript following the reviewer's suggestion.

> L350: change to 'up to 45% of total glacier area…' to make it clear that you're referring to the entire glaciated region here, and not just the proportion impacted on each individual surge-type glacier

We have fixed this using the reviewe'rs suggestion.

> L372: seem to be missing some words here? 'in mass balance a single…' doesn't make sense as written

We have rephrased this sentence.

---

## Author Response (AR3)

**Response to Technical Corrections**

The authors would like to thank the Editor for his careful reading and constructive comments on our last submission.

We apologize for making it harder for the Editor to parse our previous rebuttal letter. It appeared to us a complete rebuttal letter with responses to all questions from both reviewers and the Editor would be the best choice.

We now reply in more details to each of the technical corrections from the Editor.

> As an answer to my previous comment "L426. I would insist here on the need for high temporal resolution data. Right now this is not the case in the Hugonnet et al. dataset because ASTER acquisitions were scarce and their uncertainties rather high", you simply wrote in the rebuttal letter " Following the reviewers' ". It seems that part of your answer was deleted. Can you provide the full sentence in a brief response?

We apologize for the incomplete answer in our previous rebuttal letter. Following the Editor's comments on L426, we have now reworked the sentence which now reads: "The impact of surging over more localized scales thus need to be further quantified. This requires the thorough study of a greater number of higher temporal resolution datasets to cover the entire surge cycle of a significant glacier sample and provide low-uncertainty mass balance estimates."

> L442. In my previous review, I suggested the replacement of "than" by "that". You made this replacement but for the wrong line! So now an error between "than"/"that" needs to be corrected L442 and L443.

We have fixed this.